# MSPipe: Minimal Staleness Pipeline for Efficient Temporal GNN Training

## Abstract

Temporal graph neural networks (TGNNs) have demonstrated exceptional performance in modeling interactions on dynamic graphs. However, the adoption of memory modules in state-of-the-art TGNNs introduces significant overhead, leading to performance bottlenecks during training. This paper presents MSPipe, a minimal staleness pipeline design for maximizing training throughput of memory-based TGNNs, tailored to maintain model accuracy and reduce resource contention. Our design addresses the unique challenges associated with fetching and updating memory modules in TGNNs. We propose an online pipeline scheduling algorithm that strategically breaks temporal dependencies between iterations with minimal staleness and delays memory fetching (for obtaining fresher memory vectors) without stalling the GNN training stage or causing resource contention. We further design a staleness mitigation mechanism to improve training convergence and model accuracy. We provide convergence analysis and demonstrate that MSPipe retains the same convergence rate as vanilla sampling-based GNN training. Our experiments show that MSPipe achieves up to $2.45\times$ speed-up without sacrificing accuracy, making it a promising solution for efficient TGNN training. The implementation (anonymous) for our paper can be found at https://anonymous.4open.science/r/MSPipe/.

## 1 Introduction

Many real-world graphs are inherently dynamic with nodes and edges continuously evolving over time. For example, a temporal graph of a social network captures the changing patterns of connections between individuals, while a temporal user-item graph can represent the changing preferences of users in a recommendation system. Previous attempts to model these dynamic systems have relied on static graph representations that fail to account for their temporal nature Zhang et al. (2019; 2020); Nguyen et al. (2018). Recently, temporal graph neural networks (TGNNs) are designed to incorporate time-related information, learn both structural and temporal dependencies and enable more accurate and comprehensive modeling of dynamic graphs Rossi et al. (2021); Wang et al. (2021); Kumar et al. (2019); Trivedi et al. (2019); Xu et al. (2020); Zhang et al. (2023); Cong et al. (2023).

Among the existing TGNN models, memory-based TGNNs, such as TGN Rossi et al. (2021), APAN Wang et al. (2021), JODIE Kumar et al. (2019), DyRep Trivedi et al. (2019) and TIGER Zhang et al. (2023), have achieved state-of-the-art performance on a variety of tasks Poursafaei et al. (2022), notably link prediction and node classification. Their promising performance can be attributed to the node memory module, which stores time-aware representations, allowing them to capture intricate long-term information of each node. In each training iteration of these models, the memory vectors of nodes within the sampled subgraphs are loaded and fed into GNN training together with node features (Fig. 1); the updated representations are written back to the memory module along with the current batch of the timed events when this iteration of GNN training is done.

Despite their impressive performance, it remains challenging to train memory-based TGNN at scale, because of the temporal dependency induced by the nature of the memory module. The memory module behaves like a recursive filtering mechanism that iteratively filters and distills information of historical events into the memory state. Consequently, respecting the temporal dependency incurs a substantial overhead in memory-based TGNN training (up to 36.1% of the execution time of one training iteration) but is important for maintaining model performance. Specifically, the temporal dependency of the memory module is manifested in memory fetch and update operations. **First**, the

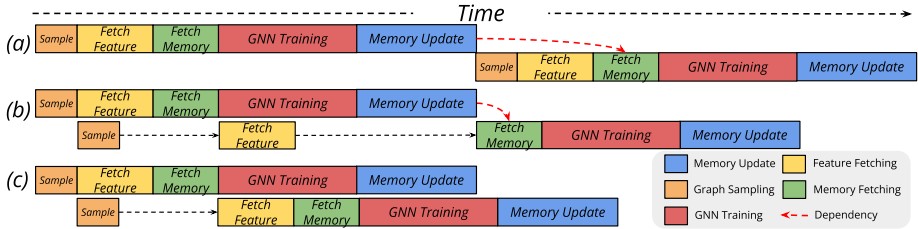

Figure 1: Memory-based TGNN Training. (a) represents the general training scheme; (b) shows the pre-sampling and pre-fetching optimization; (c) is the case of breaking the dependency.

latest memory state of a node cannot be fetched until the update of the node memory module in the last iteration has been completed. This dependency is illustrated by the red arrow in Fig. 1, indicating that subsequent iterations fetch the most recently updated node memory from previous iterations. **Next**, to avoid information leakage, updated memory of a node with a graph event in the current batch cannot be utilized before training the model using the same event Rossi et al. (2021). As a result, the memory update can only be applied at the end of each training iteration. When applying optimizations from previous works Kaler et al. (2022); Zheng et al. (2022), memory fetching in the next training iteration has to wait until the memory update is finished as shown in Fig. 1(b), diminishing the training efficiency. The detailed temporal dependencies originating from the memory module between iterations are depicted in Fig. 2. These temporal dependencies on the memory fetch and update execution orders limit the chances for parallel execution and lower GPU utilization in training TGNNs, posing a significant challenge in effectively scaling TGNN learning to large datasets. In contrast, the remarkable success of recent DNN models relies on efficient parallelism for maximal utilization of distributed hardware resources and datasets Shoeybi et al. (2019); Rasley et al. (2020). Therefore, there is a pressing need for a parallel execution scheme that enables more efficient and scalable distributed TGNN training.

There exists a line of research Wan et al. (2022); Peng et al. (2022); Zheng et al. (2022); Kaler et al. (2022); Gandhi & Iyer (2021) on optimizing the training of static GNNs. However, the temporal dependencies discussed above are unique to TGNN because of the memory module. The existing works fail to handle such temporal dependencies and are therefore ineffective for TGNN training. Moreover, prior studies on optimizing TGNN systems are limited Wang & Mendis (2023); Zhou et al. (2022a), which all focus on accelerating inference speed rather than training throughput. While previous works have analyzed the convergence rate of static GNN training Chen et al. (2017); Cong et al. (2020; 2021), the ramifications of violating temporal dependencies have not yet been explored.

To fill these gaps, we propose a novel framework that leverages a minimal staleness bound to accelerate TGNN training with theoretical guarantees. The main contributions are as follows:

• We propose a formulation for the TGNN training pipeline, by analyzing the initiation and completion times across training stages. This formulation decomposes TGNN training into different stages, facilitating a comprehensive analysis of training bottlenecks. Using this formulation, we conduct thorough profiling of distributed memory-based TGNN training, illustrating the opportunity to optimize the bottlenecks caused by the memory module and its temporal dependencies.

• Tackling the temporal dependencies, we propose a TGNN training framework, MSPipe, consisting of two key designs: (1) We break the temporal dependencies by introducing staleness in the memory module. This is achieved through a minimal staleness algorithm that determines a minimal staleness bound and effectively schedules the training pipeline accordingly, to maximize training throughput while maintaining model accuracy. (2) We propose a lightweight staleness mitigation method that leverages the memories of recently updated nodes with the highest similarity, which effectively reduces the staleness error.

• We provide a theoretical convergence analysis, demonstrating that MSPipe does not sacrifice convergence speed and the asymptotic convergence rate of our method is the same as vanilla memory-based TGNN training (without staleness).

• We comprehensively evaluate the performance of MSPipe through experiments. MSPipe outperforms existing state-of-the-art frameworks (TGL Zhou et al. (2022b) and TGL with optimizations from SALIENT Kaler et al. (2022)), achieving up to 2.45× speed-up and 83.6% scaling efficiency without accuracy loss.

## 2 BACKGROUND AND RELATED WORK

**Temporal Graph Neural Network.** Among the variety of TGNNs, memory-based TGNNs achieve state-of-the-art accuracy in modeling temporal dynamics in graph-structured data Rossi et al. (2021); Xu et al. (2020); Trivedi et al. (2019); Kumar et al. (2019); Poursafaei et al. (2022); Sankar et al. (2020); Zhang et al. (2023); Cong et al. (2023). Memory-based TGNNs maintain a node memory vector $S_v$ for each node $v$ in the dynamic graph that memorizes long-term dependencies. The memory update and training paradigms can be formulated as:

$$\overline{m}_v^i = agg(m_v^i, m_u^i \mid u \in N(v)) \tag{1}$$

$$S_v^i = f(S_v^{i-1}, \overline{m}_v^i) \tag{2}$$

$$h_v^i = \phi(g(S_v^i, S_u^i \mid u \in N(v)) \tag{3}$$

where $m_v^i$ represents a message generated by a graph event related to $v$ that occurs at training iteration $i$, $S_v^i$ is the memory vector and $h_v^i$ is the embedding of node $v$ in iteration $i$. The messages of node $v$'s 1-hop neighbors $N(v)$ are combined with $m_v^i$ using an aggregator function $agg$ (e.g., mean aggregation) to form $\overline{m}_v^i$, which are applied in the memory update function $f(\cdot)$ (e.g., an RNN) to update $S_v^i$.

Then the memory aggregator $g(\cdot)$ gathers the memory vector and passes it to the update function $\phi(\cdot)$, which is a single layer perceptron $\sigma(W_i)$ where $\sigma(\cdot)$ is a non-linear activation function and $W_i$ is a weight matrix. Note that all the above operations are executed in GPU and the updated memory vectors $S_v^i$ will be written back to the memory storage in CPU main memory. The detailed training workflow is illustrated in Fig. 2. The key design differences of various TGNN models lie in $f(\cdot)$

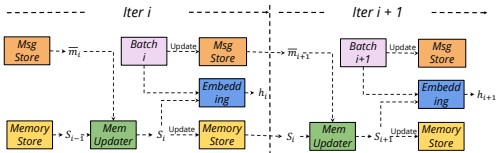

Figure 2: Memory-based TGNN Training Stages

and $g(\cdot)$ functions. Due to the space limit, detailed discussion of the TGNN models can be found in Appendix B.

**Optimizations for GNNs.** There have been recent studies on accelerating the inference speed of TGNN Zhou et al. (2022a); Wang & Mendis (2023). Sampling-based mini-batch training has become the norm for static GNN and TGNN training Gandhi & Iyer (2021); Hamilton et al. (2017); Waleffe et al. (2023); Yang et al. (2022); Ying et al. (2018), which samples a subset of neighbors of target nodes to generate a subgraph, as input to GNN training. The bottlenecks mainly lie in subgraph sampling and feature fetching due to the neighbor explosion problem Chen et al. (2018); Yan et al. (2018). ByteGNN Zheng et al. (2022) and SALIENT Kaler et al. (2022) adopt pre-sampling and pre-fetching to hide sampling and feature fetching overhead in multi-layer static GNN training. These optimizations may not resolve the bottleneck of TGNN training: Maintaining node memories in sequential order presents inevitable overhead in memory-based TGNN training when only lightweight sampling and feature fetching are needed for a single TGNN layer Rossi et al. (2021).

**Asynchronous Distributed Training** A number of works advocate asynchronous training of DNN and static GNN models, which introduces staleness in model parameter learning. For distributed DNN training, PipeSGD Li et al. (2018), SAPipe Chen et al. (2022), Hogwild Recht et al. (2011), SSP Ho et al. (2013) and Dai et al. (2018) adopt stale weight gradients on large model parameters to eliminate communication overhead, while GNN models typically have much smaller sizes. For static GNN training, PipeGCN Wan et al. (2022) and Sancus Peng et al. (2022) overlap model computation with communication in full-graph training, and introduce staleness in node embeddings. Although these methods are effective in training multi-layer static GNNs, their effect is limited when applied to memory-based TGNNs, from three aspects: **1)** they focus on full graph training and apply staleness between multiple GNN layers to overlap the significant communication overhead with computation. In TGNN training, the communication overhead is relatively small due to subgraph sampling and the presence of only one GNN layer. Therefore, the optimizations for full-graph training are not suitable. **2)** all previous GNN training frameworks simply introduce a pre-defined staleness bound without explicitly analyzing the relationship between model quality and training throughput, potentially leading to sub-optimal parallelization solutions; **3)** they focus on addressing the logical dependency between feature fetching and model training within each training iteration, while performance bottleneck lies more on temporal dependency caused by memory fetching and updating in TGNN training. To our best knowledge, we present the first comprehensive approach to address the bottlenecks arising from

| Dataset | Sample | Fetch feature | Fetch memory | Train GNN | Update memory |
|---------|--------|---------------|--------------|-----------|---------------|
| REDDIT Kumar et al. (2019) | 9.5% | 12.6% | 5.7% | 46.9% | 25.3% |
| WIKI Kumar et al. (2019) | 6.6% | 5.8% | 5.8% | 51.5% | 30.3% |
| MOOC Kumar et al. (2019) | 9.7% | 3.0% | 2.5% | 53.1% | 31.7% |
| LASTFM Kumar et al. (2019) | 11.5% | 9.1% | 8.5% | 43.0% | 26.8% |
| GDELT Zhou et al. (2022b) | 17.6% | 12.8% | 10.5% | 37.5% | 21.6% |

Table 1: Training time breakdown of TGN model. Profiling setup and other models' statistics are in Appendix D.

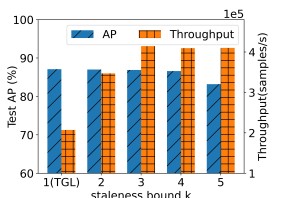

Figure 3: Training throughput and model accuracy under different staleness bounds

the memory module in TGNNs and determine the minimal staleness bound that enables efficient and scalable training of memory-based TGNNs, with corresponding theoretical convergence proofs.

## 3 MSPIPE FRAMEWORK

We design a stall-free minimal-staleness scheduling system for TGNN training, MSPipe (Fig. 1(c)). We first identify the bottleneck and temporal dependencies triggered by the memory module. To accelerate training, we pipeline data preprocessing, memory module fetching/updating and GNN training across multiple iterations. The minimal number of staleness iterations without causing pipeline stagnation is optimally decided and an online scheduling algorithm is designed to control memory staleness and avoid resource contention. To further mitigate memory staleness, we propose a lightweight similarity-based memory update module to obtain fresher information.

### 3.1 MSPIPE MECHANISM

**Significant memory operation overhead and temporal dependencies.** We consider a common 5-stage abstraction of memory-based TGNN training, i.e., graph sampling, feature fetching, memory fetching, GNN training and memory update. We conduct detailed profiling of the execution time of each stage, with time breakdown shown in Table 1. Memory operations incur substantial overhead (up to 36.1% of the total execution time of one training iteration), while sampling and feature fetching do not (due to the 1-layer TGNN structure). As shown in Fig. 1(a), memory fetching depends on memory vectors updated at the end of the previous iteration, and has to wait for the relatively long TGNN training and memory updating to finish (Fig. 1(b)).

**Pipline mechanism.** A natural design to accelerate the training process involves decoupling the temporal dependency between the memory update stage in one training iteration and the memory fetching stage in the subsequent iteration, by leveraging stale memory vectors in the latter. Fig. 1(c) provides a high-level overview of the training pipeline, where computation (e.g., GNN training) is parallelized with fragmented I/O operations encompassing feature fetching, memory fetching, and memory update. The advanced memory fetching stage introduces a certain degree of staleness into the node memory module, causing the TGNN model to receive outdated input. Mathematically, MSPipe's training can be formulated as follows, which are modified from Eqn. 2 and Eqn. 3:

$$\tilde{S}_v^i = f(\tilde{S}_v^{i-k}, \overline{m}_v^i), h_v^i = \phi(g(\tilde{S}_v^i, \tilde{S}_u^i \mid u \in N(v)) \tag{4}$$

where $\tilde{S}_v^i$ represents the memory vector of node $v$ in training iteration $i$ updated based on stale memory vector in iteration $i - k$, and $h_v^i$ is the embedding of node $v$. MSPipe uses the memory vector from $k$ iterations before the current iteration to generate messages and train the model.

In the example pipeline in Fig. 4, we have staleness bound $k = 2, 4$, indicating that memory fetching retrieves memory vectors updated two and four iterations before, respectively. While a larger staleness bound can enhance throughput, it may have detrimental effects on model convergence. Previous static GNN frameworks Wan et al. (2022); Peng et al. (2022) all rely on a intuitively predefined staleness bound. We argue that randomly selecting a staleness bound $k$, is inadequate and may lead to delayed execution of the GNN training stage or unnecessarily high staleness. To support our argument, we conduct experiments on the LastFM dataset, training TGN models. As depicted in Fig. 3, applying the smallest staleness bound (e.g., $k = 2$) leads to degradation in training throughput, while employing a larger staleness bound (e.g., $k = 5$) impacted model accuracy. To address this, we introduce a pipeline scheduling policy that determines the minimal staleness bound that maximizes system throughput without affecting model convergence.

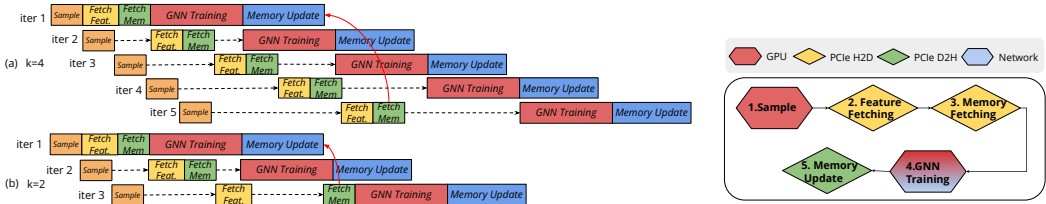

Figure 4: Pipeline execution. Dashed arrow represents the bubble time. Red arrow denotes memory fetching to retrieve memory vectors updated $k$ iterations before.

Figure 5: Resource requirement of different stages.

## 3.2 STALL-FREE MINIMAL-STALENESS PIPELINE SCHEDULE

To maximize TGNN training throughput, our objective is to enable the GPU to seamlessly perform computation without waiting for data preparation, as depicted in Fig. 4(a). We seek to determine the minimal staleness iteration number $k$ and perform resource-aware online pipeline stage scheduling, which enables maximum speed-up without stalling the GNN training stage, while ensuring model convergence. To accurately model resource contention, we analyze the resource requirements for different stages. Fig. 5 demonstrates that feature fetching and memory fetching contend for the copy engine and PCIe resources during the copy operation from host to device. However, no contension are encountered during the memory update stage, as it involves a copy operation from device to host Choquette & Gandhi (2020). Additionally, we adopt a GPU sampler with restricted GPU resource allocation to avoid competition with the GNN training stage.

**Minimal-staleness bound k.** Let $b_i^{(j)}$ and $e_i^{(j)}$ denote the start time and end time of stage $j$ in iteration $i$. $\tau^{(j)}$ is the execution time of stage $j$. $b_i^{(j)}$ and $e_i^{(j)}$ can be computed as

$$
b_i^{(j)} = \begin{cases} b_{i-1}^{(j)} + \tau^{(j)} & j = 1 \\ \max(b_i^{(j-1)} + \tau^{(j-1)}, b_{i-1}^{(j+1)} + \tau^{(j+1)}) & j = 2 \\ \max(b_i^{(j-1)} + \tau^{(j-1)}, b_{i-1}^{(j)} + \tau^{(j)}) & j \in [3,5] \end{cases}, \quad e_i^{(j)} = b_i^{(j)} + \tau^{(j)}, j \in [1,5]
$$

(5)

where $\tau^{(j)}$ can be collected in a few iterations of profiling. Eqn. 5 ensures sequential execution of the stages in each training iteration, which prohibits simultaneous execution of the same stage from different iterations and resource competition among different stages. Specifically, feature fetching (stage 2) competes for PCIe and copy engine resources with memory fetching (stage 3) in the last iteration. Consequently, in MSPipe, feature fetching cannot commence until the memory fetching from the previous iteration and the sampling stage from the current iteration have both been completed, as illustrated in Fig. 4. Moreover, during synchronous training, different batches cannot be computed simultaneously by the TGNN (stage 4) and memory vectors from different iterations must be updated sequentially (stage 5) to prevent write conflicts.

To maximize training throughput with the least impact on model accuracy, we seek to fetch the most up-to-date memory vectors that are $k_i$ iterations before the current iteration $i$ without causing resource contention or pipeline stalling. To achieve this, we find the minimal $k_i$ as in the following linear program:

$$
\text{minimize} \quad k_i
$$
$$
\text{subject to} \quad e_{i-k_i}^{(j_{\text{upd}})} \geq b_i^{(j_{\text{feat}})} + \tau^{(j_{\text{feat}})}, \ e_{i-k_i}^{(j_{\text{upd}})} \leq b_i^{(j_{\text{train}})} - \tau^{(j_{\text{mem}})}, \ k_i \leq k_{max}
$$
$$
1 \leq k_i < i, i = 1, ..., E, i - k_i \geq 0, j_{\text{feat}} = 2, j_{\text{mem}} = 3, j_{\text{train}} = 4, j_{\text{upd}} = 5
$$

Here $E$ denotes the total number of iterations in a training epoch. There are 3 constraints: **1)** The first ensures that the memory update for the $i - k_i$th iteration is finished before memory fetching in the $i$th iteration. **2)** The second guarantees that delaying the memory fetching stage does not stall the subsequent GNN training stage, which enables incessant execution of GNN training stages on the GPU. **3)** The third constraint introduces an upper bound for the number of staled iterations, which is based on an observation: *During each iteration, the memory module updates only a small subset of nodes' memory vectors.* Consequently, it is only the memory vectors of these specific nodes that become stale when they are fetched prior to the memory update stage. In Fig. 6, it can be observed

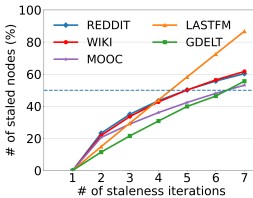

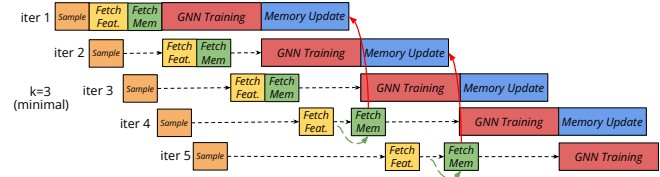

Figure 6: Percentage of nodes that use staled memory vectors under different numbers of staleness iterations

Figure 7: Resource-aware online schedule. Red arrow denotes memory fetching to retrieve memory vectors updated $k$ iterations before. Green dashed arrow represents the delay time.

that the percentage of stale nodes increases with larger staleness iterations and we choose an upper bound $k_{max}$ to ensure that the percentage of stale nodes will not exceed 50%. By iterating through each iteration, the above problem can be solved in $O(E)$.

**Resource-aware online pipeline schedule.** Once we have determined the minimal staleness iteration number $k_i$, we can schedule the training pipeline by deciding the start time $b_i^{(j)}$ of each stage. This scheduling problem can be modeled as a variant of the "bounded buffer problem" in producer-consumer systems Mehmood et al. (2011), where the buffer length $k$ represents the number of staled iterations, the memory update stage serves as a slow consumer, and the memory fetching stage acts as a fast producer. To ensure efficient training, the scheduler ensures that the training stages from different iterations do not compete for the same hardware resource and strictly adhere to a sequential execution order, as illustrated in Fig. 7. By leveraging the minimal staleness iteration numbers $k_i$, the scheduler monitors the staleness state of each iteration and defers the memory fetching stage until the minimal staleness condition is satisfied, ensuring that subsequent GNN training stages are not impeded to maximize training throughput. The detailed pseudocode can be found in Appendix D.3.

### 3.3 SIMILARITY-BASED STALENESS MITIGATION

As a node's memory should be updated upon occurrence of events related to the node, stationary memory state over a certain period of time would result in stale representations Rossi et al. (2021); Kumar et al. (2019). MSPipe may aggravate this problem although minimal staleness is introduced. To improve model convergence and accuracy with MSPipe, we further propose a staleness mitigation strategy by aggregating memories of recently active nodes with the highest similarity, which are considered to have similar and fresher temporal representations, to update the stale memory of a node. When node $v$'s memory has not been updated for time $\Delta t$, longer than a

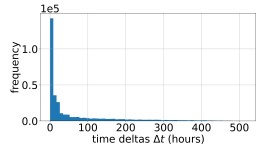

Figure 8: Distribution of $\Delta t$ in WIKI dataset

threshold $t_{thr}$, we update the stale memory of the node, $\tilde{S}_v^{i-k}$, by combining it with the averaged memory of a set $N$ of most similar and active nodes. An active node is defined to be the one whose memory is fresher than that of node $v$ and $\Delta t$ is smaller than $t_{thr}$. For similarity between different nodes, we count their common neighbors which are reminiscent of the Jaccard similarity Leskovec et al. (2020). We observe that $\Delta t$ follows a power-law distribution shown in Fig. 8, which means that only a few $\Delta t$ values are much larger than the rest. We accordingly set $t_{thr}$ to $p$ quantile (e.g., 99% quantile) of the $\Delta t$ distribution to reduce staleness errors. Noted that our $\Delta t$ is the difference between the timestamp of the current event and the last updation time, which is different from observations in previous TGNN inference systems Wang & Mendis (2023); Zhou et al. (2022a). We apply the following memory staleness mitigation mechanism in the memory fetching stage:

$$\hat{S}_v^{i-k} = \lambda \tilde{S}_v^{i-k} + (1 - \lambda) \frac{\sum_{n \in N} \tilde{S}_n^{i-k}}{|N|}$$

where $\hat{S}_v^{i-k}$ is the mitigated memory vector of node $v$ at iteration $i - k$, and $\lambda$ is a hyperparameter in $[0, 1]$. The mitigated memory vector will then be fed into the memory update function:

$$\hat{S}_v^i = f(\hat{S}_v^{i-k}, \overline{m}_v^i)$$

### 3.4 THEORETICAL ANALYSIS

We analyze the convergence guarantee and convergence rate of MSPipe with respect to our bounded memory vector staleness. By carefully scheduling the pipeline and utilizing stale memory vectors,

we demonstrate that our approach incurs negligible approximation errors that can be bounded. We provide a rigorous analysis of the convergence properties of our approach, which establishes the theoretical foundation for its effectiveness in practice.

**Theorem 1 (Convergent result, informal)** *With a memory-based TGNN model, suppose that **1)** there is a bounded difference between the stale node memory vector $\tilde{S}^i$ and the exact node memory vector $S^i$ with the staleness bound $\epsilon_s$, i.e., $\|\tilde{S}^i - S^i\|_F \leq \epsilon_s$ where $\|\|_F$ is the Frobenius norm; **2)** the loss function $\mathcal{L}$ in TGNN training is bounded below and L-smooth; and **3)** the gradient of the loss function $\mathcal{L}$ is $\rho$-Lipschitz continuous. Choose step size $\eta = \min\{\frac{2}{L}, \frac{1}{\sqrt{t}}\}$. There exists a constant $D > 0$ such that:*

$$\min_{1 \leq t \leq T} \|\nabla \mathcal{L}(W_t)\|_F^2 \leq [2\mathcal{L}(W_0) - \mathcal{L}(W^*) + \rho D]\frac{1}{\sqrt{T}},$$

*where $W_0$, $W_t$ and $W^*$ are the initial, step-t and optimal model parameters, respectively.*

The formal version of Theorem 1 along with its proof can be found in Appendix A. Theorem 1 indicates that the convergence rate of MSPipe is $O(T^{-\frac{1}{2}})$, which shows that our approach maintains the same convergence rate as vanilla sampling-based GNN training methods ($O(T^{-\frac{1}{2}})$ Chen et al. (2017); Cong et al. (2020; 2021)).

## 4 EXPERIMENTS

**Testbed.** The main experiments are conducted on a machine equipped with two 64-core AMD EPYC 7H12 CPUs, 512GB DRAM, and four NVIDIA A100 GPUs (40GB), and the scalability experiments are conducted on two of such machines with 100Gbps interconnect bandwidth.

**Datasets and Models.** We evaluate MSPipe on five temporal datasets: REDDIT, WIKI, MOOC, LASTFM Kumar et al. (2019) and a large dataset GDELT Zhou et al. (2022b). On each dataset, we use the same 70%-15%-15% chronological train/validation/test set split as in previous works Xu et al. (2020); Rossi et al. (2021). More detailed information of the datasets is given in Appendix E. We train 3 state-of-the-art memory-based TGNN models, JODIE Kumar et al. (2019), TGN Rossi et al. (2021) and APAN Wang et al. (2021).

**Baselines.** We adopt **TGL** Zhou et al. (2022b), a state-of-the-art TGNN training system, as the synchronous TGNN training baseline. We run experiments using the same model hyperparameters and training settings as in TGL, which are summarized in Appendix E.2. We also implement the **Presample** (with pre-fetching features) mechanism similar to SAILENT Kaler et al. (2022) on TGL as a stricter baseline, which provides a parallel sampling and feature fetching scheme by executing them in advance. We implement **MSPipe** on PyTorch Paszke et al. (2019) and DGL Wang et al. (2019), supporting both single-machine multi-GPU and multi-machine distributed TGNN training. **MSPipe-S** is MSPipe with staleness compensation from similar neighbors with $\lambda$ set to 0.95. Noted that MSPipe does not implement the staleness mitigation mechanism by default.

### 4.1 EXPEDITED TRAINING WITH COMPARABLE ACCURACY

The results in Table 2 show that MSPipe improves the training throughput while maintaining high model accuracy. AP in the table stands for average model precision evaluated on the test set. For a more comprehensive analysis of various batch sizes, we provide detailed experiments in Appendix E.5.

**Training Throughput.** We observe that MSPipe is 1.50× to 2.45× faster than TGL, and achieves up to 104% speed-up as compared to the Presample mechanism. MSPipe obtains the best speed-up on GDELT, which can be attributed to the relatively smaller proportion of execution time devoted to the GNN training stage compared to other datasets (as shown in Table 1). This is mainly because MSPipe effectively addresses the primary bottlenecks in memory-based TGNN training by breaking temporal dependencies between iterations and ensuring uninterrupted progression of the GNN training stage, thereby enabling seamless overlap with other stages. Consequently, the total training time is predominantly determined by the uninterrupted GNN training stage. Notably, a smaller GNN training stage results in a larger speed-up, further contributing to the superior performance of MSPipe.

Table 2: Training Performance. The best (second-best) results are in **bold** (underlined). Each data point is average of 3 trials. Due to space limit, the full table with standard deviation is in Appendix.

| Model | Scheme | REDDIT | | WIKI | | MOOC | | LASTFM | | GDELT | |
|-------|--------|--------|--------|--------|--------|--------|--------|--------|--------|--------|--------|
| | | AP(%) | Speedup | AP(%) | Speedup | AP(%) | Speedup | AP(%) | Speedup | AP(%) | Speedup |
| TGN | TGL | **99.8** | 1× | **99.4** | 1× | **99.4** | 1× | 87.2 | 1× | **98.2** | 1× |
| | Presample | **99.8** | 1.16× | **99.4** | 1.12× | **99.4** | 1.16× | 87.1 | 1.36× | 98.1 | 1.32× |
| | MSPipe | **99.8** | **1.77×** | 99.1 | **1.54×** | 99.3 | **1.50×** | 86.9 | **2.00×** | **98.2** | **2.36×** |
| | MSPipe-S | **99.8** | 1.72× | 99.3 | 1.52× | **99.4** | 1.47× | **87.9** | 1.96× | **98.2** | 2.26× |
| JODIE | TGL | **99.6** | 1× | **98.4** | 1× | **98.6** | 1× | 73.0 | 1× | 98.0 | 1× |
| | Presample | **99.6** | 1.10× | **98.4** | 1.14× | **98.6** | 1.09× | 73.0 | 1.37× | 98.0 | 1.73× |
| | MSPipe | **99.6** | **1.55×** | 97.2 | **1.65×** | **98.6** | **1.50×** | 71.7 | **1.87×** | 98.1 | 2.28× |
| | MSPipe-S | **99.6** | 1.50× | 97.6 | 1.54× | **98.6** | 1.48× | **76.3** | 1.79× | **98.2** | 2.23× |
| APAN | TGL | **99.6** | 1× | **98.0** | 1× | **98.6** | 1× | 73.4 | 1× | 95.8 | 1× |
| | Presample | **99.6** | 1.38× | **98.0** | 1.06× | **98.6** | 1.30× | 73.2 | 1.49× | 95.8 | 1.71× |
| | MSPipe | **99.6** | **2.03×** | 96.4 | **1.78×** | 98.4 | **1.91×** | 72.4 | **2.37×** | 95.9 | **2.45×** |
| | MSPipe-S | **99.6** | 1.96× | 97.1 | 1.63× | **98.6** | 1.77× | **76.1** | 2.25× | **96.0** | 2.41× |

(a) REDDIT     (b) WIKI     (c) LASTFM     (d) GDELT

Figure 9: Scalability of training TGN.

**Model Accuracy.** MSPipe without staleness mitigation can already achieve comparable test average precision with TGL on all datasets, with a marginal degradation ranging from 0 to 1.6%. This can be attributed to the minimal staleness mechanism and proper pipeline scheduling in MSPipe.

**Staleness Mitigation.** With the proposed staleness mitigation mechanism, MSPipe-S consistently achieves higher average precision than MSPipe across all models and datasets. Notably, MSPipe-S achieves the same test accuracy as TGL on REDDIT and MOOC datasets, while surpassing TGL's model performance on LastFM and GDELT datasets. MSPipe-S introduces a minimal overhead of only 3.73% on average for the staleness mitigation process. This demonstrates the efficiency of the proposed mechanism in effectively mitigating staleness while maintaining high performance levels.

**Scalability.** Fig. 9 presents the training throughput with different numbers of GPUs. MSPipe achieves not only consistent speed-up but also up to 83.6% scaling efficiency on a single machine, which is computed as the ratio of the speed-up achieved by using 4 GPUs to the ideal speed-up, outperforming other baselines. We also scale TGN training on GDELT to two machines with eight GPUs in Fig. 9(d). Without explicit optimization for inter-machine communication, MSPipe still outperforms the baselines and exhibits better scalability.

## 4.2 PRESERVING CONVERGENCE RATE

To validate that MSPipe can maintain the same convergence rate as vanilla sampling-based GNN training without applying staleness ($O(T^{-\frac{1}{2}})$), we compare the training curves of all models on all datasets in Fig. 10 (the complete result can be found in Appendix E.3). We observe that MSPipe's training curves largely overlap with those of vanilla methods (TGL and Presample), verifying our theoretical results in Sec. 3.4. With staleness mitigation, MSPipe-S can achieve slightly better and more steady convergence (e.g., on WIKI and LastFM) than others.

## 4.3 STALL-FREE MINIMAL STALENESS BOUND

To further validate that MSPipe can find the minimal staleness bound without delaying the GNN training stage, we conduct a comparative analysis of accuracy and throughput between the converged staleness bound computed by MSPipe and different staleness bounds $k$. The results, depicted in Fig. 11 and Fig. 17 in Appendix E.3.3 (due to space constraints), consistently demonstrate that MSPipe achieves the highest throughput while maintaining the best accuracy compared to other

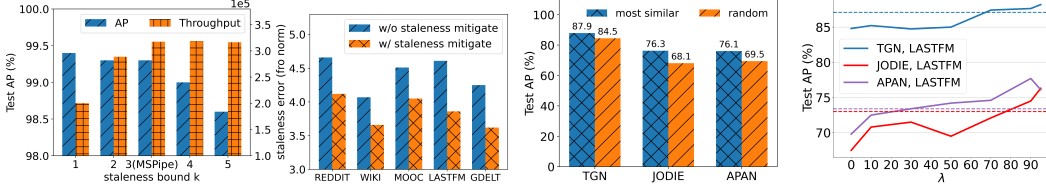

(a) REDDIT  (b) WIKI  (c) MOOC  (d) LASTFM

Figure 10: Convergence of TGN training. x-axis is the wall-clock training time, and y-axis is the test average precision.

Figure 11: Throughput and AP on different staleness bound (MOOC)

Figure 12: Staleness error comparison on TGN

Figure 13: Staleness mitigation with most similar or random nodes

Figure 14: Hyperparameter analysis

staleness bound options. Additionally, the computed minimal staleness bounds for various datasets range from 2 to 4, providing further evidence for the necessity of accurately determining the minimal staleness bound rather than relying on random selection. Note that $k = 1$ represents the baseline method of TGL without applying staleness.

## 4.4 STALENESS MITIGATION MECHANISM

**Error reduction.** To better understand the accuracy enhancement and convergence speed-up achieved by MSPipe-S, we conduct a detailed analysis of the intermediate steps involved in our staleness mitigation mechanism. Specifically, we refer to Theorem 1, where we assume the existence of a bounded difference $\epsilon_s$ between the stale node memory vector $\tilde{S}^i$ and the precise node memory vector $S^i$. To assess the effectiveness of our staleness mitigation mechanism, we compare the mitigated staleness error $\|\hat{S}^i - S^i\|_F$ obtained after applying our mechanism with the original staleness error $\|\tilde{S}^i - S^i\|_F$. As shown in Fig. 12, MSPipe-S consistently reduces the staleness error across all datasets, validating the theoretical guarantee and the effectiveness in enhancing accuracy.

**Benefit of using most-similar neighbors.** We further investigate our staleness mitigation mechanism by comparing using the most similar and active nodes for staleness mitigation with utilizing random active nodes, on the LastFM dataset. In Fig. 13, we observe that our proposed most similar mechanism leads to better model performance, while random selection would even degrade model accuracy. This could be because similar nodes have resembling representations that facilitate the stale node to obtain more updated information. The memory similarity comparison between the most similar nodes and random nodes is presented in Appendix E.3.

**Hyperparameter analysis.** We examine the effect of hyperparameter $\lambda$ on test accuracy, as depicted in Fig. 14. We find that mitigating staleness with a larger $\lambda$ ($> 0.8$) results in better model performance than TGL's results, indicating that we should retain more of the original stale memory representations and apply a small portion of mitigation from their similar ones.

## 5 CONCLUSION

We present MSPipe, an efficient and scalable pipeline scheduling framework for memory-based TGNN training that improves training throughput and maintains model accuracy. MSPipe identifies the minimal number of staleness iterations to adopt in the pipeline without causing pipeline stagnation. Given the minimal staleness, MSPipe utilizes an online scheduler to delay the memory fetching stage and prevent resource contention. MSPipe further adopts a lightweight staleness mitigation strategy to alleviate memory staleness. Extensive experiments validate that MSPipe attains significant speed-up over state-of-the-art TGNN training schemes with minimal accuracy loss.

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

## A    PROOFS

In this section, we provide the detailed proofs of the theoretical analysis.

**Lemma 1** *If $f(\cdot)$ is $\beta$-smooth, then we have,*

$$f(y) \leq f(x) + \langle \nabla f(x), y - x \rangle + \frac{\beta}{2} \|y - x\|^2$$

*Proof.*

$$|f(y) - f(x) - \langle \nabla f(x), y - x \rangle|$$
$$=| \int_0^1 \langle \nabla f(x) + t(y - x), y - x \rangle dt - \langle \nabla f(x), y - x \rangle|$$
$$\leq \int_0^1 |\langle \nabla f(x) + t(y - x) - \nabla f(x), y - x \rangle| dt$$
$$\leq \int_0^1 \|\nabla f(x) + t(y - x) - \nabla f(x)\| \cdot \|y - x\| dt$$
$$\leq \int_0^1 t\beta \|y - x\|^2 dt$$
$$=\frac{\beta}{2} \|y - x\|^2$$

**Lemma 2** *if $\mathcal{L}(\cdot)$ is $\rho$-Lipschitz smooth, then we have*

$$\|\nabla \tilde{\mathcal{L}}(W) - \nabla \mathcal{L}(W)\|_F \leq \rho \epsilon_s$$

*where $\nabla \tilde{\mathcal{L}}(W_t)$ denote the gradient when stale memoys are used.*

*Proof.* By the assumption that there is a bounded difference between the stale node memory vector $\tilde{S}_i$ and the exact node memory vector $S_i$ with the staleness bound $\epsilon_s$, we have:

$$\|S - \tilde{S}\|_F \leq \epsilon_s$$

By smoothness of $\mathcal{L}(\cdot)$, we have

$$\|\nabla \mathcal{L}(S, W) - \nabla \mathcal{L}(\tilde{S}, W)\|_F$$
$$=\|\nabla \tilde{\mathcal{L}}(W) - \nabla \mathcal{L}(W)\|_F$$
$$\leq \rho \epsilon_s$$

**Learning Algorithms.** In the $t^{th}$ step, we have

$$W_{t+1} - W_t = -\eta_t \nabla \tilde{\mathcal{L}}(W_t) \tag{6}$$

, where $\nabla \tilde{\mathcal{L}}(W_t)$ denote the gradient when stale memoys are used and $\eta_t$ is the learning rate.

By Lemma 1 and the $L_f$-smoothness of $\mathcal{L}$ , we have

$$\mathcal{L}(W_{t+1}) - \mathcal{L}(W_t) \leq \langle W_{t+1} - W_t, \nabla \mathcal{L}(W_t) \rangle + \frac{L_f}{2} \|W_{t+1} - W_t\|_F^2 \tag{7}$$

Use Eqn. 6 to substitute, we have

$$\mathcal{L}(W_{t+1}) - \mathcal{L}(W_t) \leq \underbrace{-\eta_t \langle \nabla \tilde{\mathcal{L}}(W_t), \nabla \mathcal{L}(W_t) \rangle}_{①} + \underbrace{\frac{L_f \eta_t^2}{2} \|\nabla \tilde{\mathcal{L}}(W_t)\|_F^2}_{②} \tag{8}$$

We bound the terms step by step and let $\delta_t = \nabla \tilde{\mathcal{L}}(W_t) - \nabla \mathcal{L}(W_t)$ to subsitute in Equ. 8.

First, For ① , we have

$$- \eta_t \langle \nabla \tilde{\mathcal{L}}(W_t), \nabla \mathcal{L}(W_t) \rangle$$
$$= - \eta_t \langle \delta_t + \nabla \mathcal{L}(W_t), \nabla \mathcal{L}(W_t) \rangle$$
$$= - \eta_t [\langle \delta_t, \nabla \mathcal{L}(W_t) \rangle + \|\nabla \mathcal{L}(W_t)\|_F^2]$$

For ② , we have

$$\frac{L_f \eta_t^2}{2} \|\nabla \tilde{\mathcal{L}}(W_t)\|_F^2$$
$$= \frac{L_f \eta_t^2}{2} \|\delta_t + \nabla \mathcal{L}(W_t)\|_F^2$$
$$= \frac{L_f \eta_t^2}{2} (\|\delta_t\|_F^2 + 2\langle \delta_t, \nabla \mathcal{L}(W_t) \rangle + \|\nabla \mathcal{L}(W_t)\|_F^2)$$

Combining both ① and ② together and by the choice of learning rate $\eta_t = \frac{1}{L_f}$, we have

$$\mathcal{L}(W_{t+1}) - \mathcal{L}(W_t) \leq -(\eta_t - \frac{L_f}{2}\eta_t^2)\|\nabla \mathcal{L}(W_t)\|_F^2 + \frac{L_f \eta_t^2}{2}\|\delta_t\|_F^2$$

By Lemma 2. we have $\|\delta_t\|_F^2 \leq \rho \epsilon_s$

$$\mathcal{L}(W_{t+1}) - \mathcal{L}(W_t) \leq -(\eta_t - \frac{L_f}{2}\eta_t^2)\|\nabla \mathcal{L}(W_t)\|_F^2 + \frac{L_f \eta_t^2}{2}\rho \epsilon_s \qquad (9)$$

Rearrange Eqn. 9 and let $c = \frac{L_f \rho \epsilon_s}{2}$, we have,

$$(\eta_t - \frac{L_f}{2}\eta_t^2)\|\nabla \mathcal{L}(W_t)\|_F^2 \leq \mathcal{L}(W_t) - \mathcal{L}(W_{t+1}) + \eta_t^2 c \qquad (10)$$

Telescope sum from $t = 1...T$, we have

$$\sum_{t=1}^{T}(\eta_t - \frac{L_f \eta_t^2}{2})\|\nabla \mathcal{L}(W_t)\|_F^2 \leq \mathcal{L}(W_0) - \mathcal{L}(W_T) + \sum_{t=1}^{T} \eta_t^2 c \qquad (11)$$

$$\min_{1 \leq t \leq T} \|\nabla \mathcal{L}(W_t)\|_F^2 \leq \frac{\mathcal{L}(W_0) - \mathcal{L}(W_T)}{\sum_{t=1}^{T}(\eta_t - \frac{L_f \eta_t^2}{2})} + \frac{\sum_{t=1}^{T} \eta_t^2 c}{\sum_{t=1}^{T}(\eta_t - \frac{L_f \eta_t^2}{2})} \qquad (12)$$

Substitute Equ. 12 with $\eta_t = min\{\frac{1}{\sqrt{t}}, \frac{1}{L_f}\}$ and $\mathcal{L}(W^*) \leq \mathcal{L}(W_T)$, we have

$$\min_{1 \leq t \leq T} \|\nabla \mathcal{L}(W_t)\|_F^2$$
$$\leq (2(\mathcal{L}(W_0) - \mathcal{L}(W^*)) + \frac{c}{L_f})\frac{1}{\sqrt{T}}$$
$$\leq (2(\mathcal{L}(W_0) - \mathcal{L}(W^*)) + \frac{\rho \epsilon_s}{2})\frac{1}{\sqrt{T}}$$

Therefore, the convergence rate of MSPipe is $O(T^{-\frac{1}{2}})$, which maintains the same convergence rate as vanilla sampling-based GNN training methods ($O(T^{-\frac{1}{2}})$ Chen et al. (2017); Cong et al. (2020; 2021)).

# B    MORE DISCUSSION ON THE RELATED WORK

As discussed before, the key design space of the memory-based TGNN model lies in memory updater and memory aggregator functions. JODIE Kumar et al. (2019) updates the memory using two mutually-recursive RNNs and applies MLPs to predict the future representation of a node. Similar to JODIE, TGN Rossi et al. (2021) and APAN Wang et al. (2021) use RNN as the memory update function while incorporating an attention mechanism to capture spatial and temporal information jointly. APAN further optimizes inference speed by using asynchronous propagation. A recent work TIGER Zhang et al. (2023) improves TGN by introducing an additional memory module that stores node embeddings, and proposes a restarter for warm initialization of node representations.

Moreover, some researchers focus on optimizing the inference speed of TGNN models: Zhou et al. (2022a) propose a model-architecture co-design to reduce computation complexity and external memory access. TGOpt Wang & Mendis (2023) leverages redundancies to accelerate inference of the temporal attention mechanism and the time encoder.

There are several GNN training schemes with staleness techniques, PipeGCN Wan et al. (2022) and Sancus Peng et al. (2022), as we have discussed about the difference in Sec. 2, we would like to emphasize and detail the difference between those works and MSPipe:

1. **Dependencies and Staleness**: PipeGCN Wan et al. (2022) and Sancus Peng et al. (2022) aim to eliminate inter-layer dependencies in multi-layer GNN training to enable communication-computation overlap. In contrast, MSPipe is specifically designed to tackle temporal dependencies within the memory module of TGNN training. The dependencies and staleness in TGNN training pose unique challenges that require distinct theoretical analysis and system designs.

2. **The choice of staleness bound**: Previous staleness based static GNN methods randomly choose a staleness bound for acceleration, which may lead to suboptimal system performance and affect model accuracy. MSPipe strategical decide the minimal staleness bound that can reach the highest throughput without sacrifice the model accuracy.

3. **Bottlenecks**: In full-graph training scenarios, such as PipeGCN Wan et al. (2022) and Sancus Peng et al. (2022), the main bottleneck lies in communication between graph partitions on GPUs. Due to limited GPU memory, the graph is divided into multiple parts, leading to increased communication time during full graph training. Therefore, these methods aim to optimize the communication-computation overlap to improve training throughput. In contrast, in TGNN training, the main bottleneck stems from maintaining the memory module on the CPU and the associated challenges of updating and synchronizing it with CPU storage across multiple GPUs. MSPipe focuses on addressing this specific bottleneck. Furthermore, unlike full graph training where the entire graph structure needs to be stored in the GPU, TGNN adopts a sampling-based subgraph training approach. As a result, the communication overhead in TGNN is significantly smaller compared to full graph training.

4. **Training Paradigm and Computation Patterns**: PipeGCN Wan et al. (2022) and Sancus Peng et al. (2022) are tailored for full-graph training scenarios, which differ substantially from TGNN training in terms of training paradigm, computation patterns, and communication patterns. TGNNs typically involve sample-based subgraph training, which presents unique challenges and constraints not addressed by full graph training approaches. Therefore, the full graph training works cannot support TGNN training.

5. **Multi-Layer GNNs vs Single-Layer TGNNs**: PipeGCN Wan et al. (2022) and Sancus Peng et al. (2022) lies on the assumption that the GNN have multiple layers (e.g., GCN Kipf & Welling (2016), GAT Ying et al. (2018)) and they break the dependencies among multiple layers to overlap communication with computation. While memory-based TGNNs only have one layer with a memory module Zhou et al. (2022b); Poursafaei et al. (2022); Rossi et al. (2021); Kumar et al. (2019); Wang et al. (2021), which makes their methods lose efficacy for TGNNs.

## C  TRAINING TIME BREAKDOWN

### C.1  PROFILING SETUPS

We use TGL Zhou et al. (2022b), the SOTA TGNN training framework, on a server equipped with 4 A100 GPUs for profiling, which is the same as the experiment testbed introduced in the section 4. The local batch size for the REDDIT, WIKI, MOOC, and LastFM datasets is set to 600, while for the GDELT dataset, it is set to 4000. All the breakdown statistics are averaged over 100 epochs. All these hyperparameters are the same as the experiments. We firmly believe that, by leveraging TGL's highly optimized performance, we can evaluate bottlenecks and areas for improvement, further justifying the need for our proposed MSPipe framework.

### C.2  BREAKDOWN STATISTICS OF JODIE AND APAN

We provide the training time breakdowns for the JODIE and APAN models, which reveal that memory operations, including memory fetching and updating, can account for up to 50.51% and 58.56% of the total training time, respectively. Notably, the significant overhead is primarily due to memory operations rather than the sampling and feature fetching stages, which distinguishes these models from static GNN models and the systems designed for static GNN models.

Table 3: Training time breakdown of JODIE model

| Dataset | Sample | Fetch feature | Fetch memory | Train GNN | Update memory |
|---|---|---|---|---|---|
| REDDIT Kumar et al. (2019) | 4.14% | 8.05% | 7.36% | 50.11% | 30.34% |
| WIKI Kumar et al. (2019) | 2.20% | 1.10% | 4.95% | 46.70% | 45.05% |
| MOOC Kumar et al. (2019) | 3.41% | 1.02% | 5.80% | 51.05% | 38.71% |
| LASTFM Kumar et al. (2019) | 4.29% | 1.14% | 6.19% | 44.95% | 43.43% |
| GDELT Zhou et al. (2022b) | 3.25% | 8.56% | 9.34% | 38.75% | 40.11% |

Table 4: Training time breakdown of APAN model

| Dataset | Sample | Fetch feature | Fetch memory | Train GNN | Update memory |
|---|---|---|---|---|---|
| REDDIT Kumar et al. (2019) | 12.94% | 5.75% | 15.18% | 39.14% | 27.00% |
| WIKI Kumar et al. (2019) | 6.52% | 0.87% | 9.13% | 42.61% | 40.87% |
| MOOC Kumar et al. (2019) | 10.60% | 0.83% | 8.32% | 45.11% | 35.14% |
| LASTFM Kumar et al. (2019) | 11.12% | 1.02% | 12.26% | 41.77% | 33.83% |
| GDELT Zhou et al. (2022b) | 14.34% | 3.25% | 20.31% | 23.95% | 38.15% |

### C.3  GPU SAMPLER ANALYSIS

MSPipe utilizes a GPU sampler for improved resource utilization and faster sampling and we further clarify the remarkable speedup mainly comes from our pipeline mechanism not the GPU sampler. As shown in Tab. 5, we conducted a detailed profiling of the sampling time using TGL and found that our sampler is 24.3% faster than TGL's CPU sampler for 1-hop most recent sampling, which accounts for only 3.6% of the total training time. Therefore, the performance gain is primarily attributed to our pipeline mechanism and resource-aware minimal staleness schedule but not to the acceleration of the sampler.

### C.4  WHY DOES THE MEMORY UPDATE STAGE TAKE LONGER TIME?

The memory update takes a longer time for two reasons: **1)**In a multi-GPU environment, the memory module is stored in the CPU, allowing multiple GPUs to read simultaneously but not write simultaneously to ensure consistency and avoid conflicts; **2)** our memory fetching implementation,

Table 5: Training time breakdown of TGN model.

| Dataset | Framework | Avg Epoch(s) | Sample(s) | Fetch feature (s) | Fetch memory(s) | Train GNN(s) | Update memory(s) |
|---------|-----------|--------------|-----------|-------------------|-----------------|--------------|-------------------|
| REDDIT | TGL | 7.31 | 0.69 | 0.92 | 0.42 | 3.43 | 1.85 |
| | MSPipe-NoPipe | 7.05 | 0.44 | 0.88 | 0.41 | 3.42 | 1.90 |
| WIKI | TGL | 2.41 | 0.16 | 0.14 | 0.14 | 1.24 | 0.73 |
| | MSPipe-NoPipe | 2.32 | 0.08 | 0.12 | 0.10 | 1.20 | 0.82 |
| MOOC | TGL | 4.31 | 0.42 | 0.13 | 0.11 | 2.29 | 1.37 |
| | MSPipe-NoPipe | 4.20 | 0.31 | 0.31 | 0.21 | 2.13 | 1.41 |
| LASTFM | TGL | 13.10 | 1.50 | 1.19 | 1.11 | 5.64 | 3.65 |
| | MSPipe-NoPipe | 12.64 | 1.04 | 1.20 | 1.05 | 6.12 | 3.23 |
| GDELT | TGL | 645.46 | 113.62 | 82.39 | 67.62 | 242.61 | 139.22 |
| | MSPipe-NoPipe | 626.09 | 94.26 | 85.20 | 69.21 | 240.99 | 136.43 |

aligns with TGL, utilizes non-blocking memory copy APIs for efficient transfer of memory vectors from CPU to GPU with pinned memory. However, the lack of a non-blocking API equivalent for *tensor.cpu()* can impact performance.

# D  IMPLEMENTATION DETAILS

## D.1  ALGORITHM DETAILS

We clarify that $\tau$ is the execution time of different stages, which can be collected in a few iterations of the profiling. The $\tau$ and the staleness $k_i$ can be pre-calculated for all the graph data, which can be reused for future training. It's simple and efficient to do the profiling, pre-calculation, and training with our open-source code provided in the anonymous link.

In the case of stages such as the GNN computation stage, the execution time is likely to be dependent on the number of sampled nodes or edges. This quantity not only varies across different batches but also depends on the underlying graph structure. While the training time of a static GNN can differ due to varying numbers of neighbors for each node and the utilization of random sampling, memory-based TGNNs typically employ a fixed-size neighbor sampling approach using the most recent temporal sampler. Specifically, the sampler selects a fixed number of the most recently observed neighbors to construct the subgraph. Consequently, as the timestamp increases, the number of neighboring nodes grows, and it becomes more stable, governed by the maximum number of neighbors per node constraint. Through our profiling analysis, we observed that the number of nodes in the subgraph converges after approximately 10-20 iterations, allowing the average execution time to effectively represent the true execution time.

## D.2  MULTI-GPU SERVER IMPLEMENTATION

We have provided a brief description of how MSPipe works in multi-GPU servers at Sec. 2 and Sec. 3.1 and we have provided the implementation with the anonymous link in the abstract. We will give you a more detailed analysis of the implementation details here: The graph storage is implemented with NVIDIA UVA so each GPU worker retrieves a local batch of events and performs the sampling process on GPU to generate sub-graphs. The memory module is stored in the CPU's main memory without replication to ensure consistency and exhibit the ability to store large graphs. Noted that, except for the GPU sample, the other stages align with TGL. Here is a step-by-step overview:

1. Each GPU worker retrieves a local batch of events and performs the sampling process on the GPU to generate sub-graphs.

2. Fetches the required features and node memory vectors from the CPU to the GPU for the subgraphs.

3. Performs TGNN forward and backward computations on each GPU. MSPipe implements Data Parallel training similar to TGL.

4. The memory module is stored in the CPU's main memory without replication to ensure consistency. Each GPU transfers the updated memory vectors to the CPU and updates the corresponding elements, which ensures that the memory module remains consistent across all GPUs.

### D.3 STALL-FREE MINIMAL STALENESS SCHEDULING

We propose a resource-aware online scheduling algorithm to decide the starting time of stages in each training iteration, as given in Alg. 1

---

**Algorithm 1** Online Scheduling for TGNN training pipeline

---

1: **Input:** $E$ batches of events $\mathcal{B}_i$, Graph $\mathcal{G}$, minimum staleness iteration number $k_i$
2: **Global:** $i_{\text{upd}} \leftarrow 0$            ▷ the latest iteration whose memory update is done
3: **for** $i \in 1, 2, ..., E$ in parallel **do**
4:      **if** $lock(sample\_lock)$ **then**
5:          $\mathcal{G}_{\text{sub}} \leftarrow Sample(\mathcal{G}, \mathcal{B}_i)$            ▷ sample subgraph $\mathcal{G}_{\text{sub}}$ using a batch of events
6:      **if** $lock(feature\_lock ~\&~ pcie\_lock)$ **then**
7:          $fetch\_feature(\mathcal{G}_{\text{sub}})$            ▷ feature fetching for the subgraphs
8:      **if** $lock(memory\_lock ~\&~ pcie\_lock)$ **then**
9:          **while** $i - i_{\text{upd}} > k_i$ **do**
10:              $wait()$    ▷ delay memory fetching until staleness iteration number is smaller than $k_i$
11:          $fetch\_memory(\mathcal{G}_{\text{sub}})$           ▷ transfer memory vectors for the subgraphs
12:      **if** $lock(gnn\_lock)$ **then**
13:          $GNN(\mathcal{G}_{\text{sub}})$            ▷ train the GNN model using the subgraphs
14:      **if** $lock(update\_lock)$ **then**
15:          $update\_mem(\mathcal{G}_{\text{sub}}, \mathcal{B}_i)$ ▷ generate new memory vectors and write back to CPU storage
16:          $i_{\text{upd}} \leftarrow i$            ▷ update the last iteration with memory update done

---

To enable asynchronous and parallel execution of the stages, we utilize a thread pool and a CUDA stream pool. Each batch of data is assigned an exclusive thread and stream from the respective pools, enabling concurrent processing of multiple batches. Dedicated locks for each stage are used to resolve resource contention and enforce sequential execution (Eqn. 5). Fig. 7 provides a schematic illustration of our online scheduling. The schedule of the memory fetching stage ensures the minimal staleness iteration requirement (Lines 8-11). As illustrated in Fig. 7, the scheduling effectively fills the bubble time while minimizing staleness and avoiding resource competence. At the end of each training iteration, new memory vectors are generated based on the staled historical memories and events in the current batch (Line 15). Finally, the latest iteration whose memory update stage has been completed is recorded, enabling other parallel threads that run other training iterations to track (Line 16). Note that the first few iterations before iteration $k$ will act as a warmup, which means they will not wait for the memory update $k$ iterations before.

## E FULL EXPERIMENTS

We first provide the details of the experiments and discuss the experiment setting. Then we provide the full version of the experiment results, including the accuracy and throughput speedup, the convergence of the JODIE and APAN model, the distribution of $\Delta t$ in remaining datasets, and the analysis of node memory similarity.

### E.1 DETAILS OF THE EXPERIMENTS

**Datasets.** This paper employs several datasets, each with its unique properties and characteristics. The Reddit dataset captures the posting behavior of users on subreddits over one month, and the link feature is extracted through the conversion of post text into a feature vector. The Wikipedia dataset records the editing behavior of users on Wikipedia pages over a month, and the link feature is extracted through the conversion of the edit text into a 172-dimensional Linguistic Inquiry and Word Count (LIWC) feature vector. The MOOC dataset captures the online learning behavior of students in a MOOC course while the LastFM dataset contains information about which songs were

Table 6: The detailed statistics of the datasets. $|d_v|$ and $|d_e|$ show the dimensions of node features and edge features, respectively. Random means we use randomized features

| Dataset | $|V|$ | $|E|$ | $|d_v|$ | $|d_e|$ | Node Features | Link Features | Duration |
|---------|-----|-----|-------|-------|---------------|---------------|----------|
| Reddit Kumar et al. (2019) | 10,984 | 672,447 | 0 | 172 | No | Yes | 1 month |
| WIKI Kumar et al. (2019) | 9,227 | 157,474 | 0 | 172 | No | Yes | 1 month |
| MOOC Kumar et al. (2019) | 7,144 | 411,749 | 0 | 128 | No | Random | 17 months |
| LastFM Kumar et al. (2019) | 1,980 | 1,293,103 | 0 | 128 | No | Random | 1 month |
| GDELT Zhou et al. (2022b) | 16,682 | 1,912,909 | 413 | 186 | Yes | Yes | 5 years |

listened to by which users over one month. The GDELT dataset is a Temporal Knowledge Graph that records global events in multiple languages every 15 minutes, which covers events from 2016 to 2020 and consists of homogeneous dynamic graphs with nodes representing actors and temporal edges representing point-time events. Furthermore, it is important to highlight that *TGNN training employs graph edges as training samples*, in contrast to static GNN training, which utilizes nodes as training samples. All the datasets are downloaded from the link in TGL Zhou et al. (2022b) repository.

## E.2  EXPERIMENT SETTINGS

The implementations of TGN, JODIE, and APAN are modified from TGL Zhou et al. (2022b) for better modularity and readability. The implementations from TGL can achieve better accuracy than these models' original implementation. To ensure a fair comparison, we used the same default hyperparameters as TGL, including a learning rate of 0.0001, a local batch size of 600 (4000 for the GDELT dataset), and hidden dimensions and memory dimensions of 100. We train each dataset for 100 epochs, except for GDELT, which was trained in 10 epochs. We sampled the 10 most recent 1-hop neighbors for all datasets and constructed mini-batches with an equal number of positive and negative node pairs for sampling and subgraph construction during training and evaluation.

## E.3  FULL VERSION OF THE EXPERIMENT RESULTS

### E.3.1  ACCURACY AND THROUGHPUT SPEEDUP.

We include the missing standard deviations and a higher precision for the average precision of the evaluation in Sec. 4.1. As shown in Table 7, we can see that MSPipe without staleness mitigation can already achieve the same or even slightly better test average precision with TGL on all datasets, with up to 2.45× speedup than TGL.

Table 7: Training Performance. The best and second-best results are emphasized in **bold** and underlined. The AP difference smaller than 0.1% will be considered the same.

| Model | Dataset | REDDIT | | WIKI | | MOOC | | LASTFM | | GDELT | |
|-------|---------|--------|--------|------|--------|------|--------|--------|--------|-------|--------|
| | | AP(%) | Speedup | AP(%) | Speedup | AP(%) | Speedup | AP(%) | Speedup | AP(%) | Speedup |
| TGN | TGL | **99.82(0.03)** | 1× | **99.43(0.03)** | 1× | **99.42(0.03)** | 1× | 87.21(1.90) | 1× | 98.23(0.05) | 1× |
| | Presample | 99.80(0.03) | 1.16× | **99.43(0.03)** | 1.12× | **99.40(0.03)** | 1.16× | 87.12(1.51) | 1.36× | 98.18(0.05) | 1.32× |
| | MSPipe | 99.81(0.03) | **1.77×** | 99.14(0.03) | **1.54×** | 99.32(0.03) | **1.50×** | 86.93(0.89) | **2.00×** | **98.25(0.06)** | **2.36×** |
| | MSPipe-S | **99.82(0.03)** | 1.72× | 99.39(0.03) | 1.52× | **99.48(0.03)** | 1.47× | **87.93(1.26)** | 1.96× | **98.29(0.04)** | 2.26× |
| JODIE | TGL | **99.63(0.02)** | 1× | **98.40(0.03)** | 1× | **98.64(0.01)** | 1× | 73.04(2.89) | 1× | 98.01(0.07) | 1× |
| | Presample | **99.62(0.03)** | 1.10× | **98.41(0.03)** | 1.14× | **98.61(0.03)** | 1.09× | 72.96(2.68) | 1.37× | 98.04(0.05) | 1.73× |
| | MSPipe | **99.62(0.02)** | **1.55×** | 97.24(0.02) | **1.65×** | **98.63(0.02)** | **1.50×** | 71.7(2.84) | **1.87×** | 98.12(0.08) | **2.28×** |
| | MSPipe-S | **99.63(0.02)** | 1.50× | 97.61(0.02) | 1.54× | **98.66(0.02)** | 1.48× | **76.32(2.45)** | 1.79× | **98.23(0.05)** | 2.23× |
| APAN | TGL | **99.62(0.03)** | 1× | **98.01(0.03)** | 1× | **98.60(0.03)** | 1× | 73.37(1.59) | 1× | 95.80(0.02) | 1× |
| | Presample | 99.65(0.02) | 1.38× | **98.03(0.03)** | 1.06× | **98.62(0.03)** | 1.30× | 73.24(1.70) | 1.49× | 95.83(0.04) | 1.71× |
| | MSPipe | 99.63(0.03) | **2.03×** | 96.43(0.04) | **1.78×** | 98.38(0.02) | **1.91×** | 72.41(1.21) | **2.37×** | **95.94(0.03)** | **2.45×** |
| | MSPipe-S | 99.64(0.03) | 1.96× | 97.12(0.03) | 1.63× | **98.64(0.03)** | 1.77× | **76.08(1.42)** | 2.19× | 96.02(0.03) | 2.41× |

**The superior AP in LastFM.** The reasons why our staleness mitigation strategy outperforms the AP of the baseline TGL in the LastFM dataset is due to the unique characteristics of the LastFM datasets:

• The LastFM dataset exhibits a larger average time gap ($\frac{t_{max}-t_{min}}{E}$, where $t_{max}$ and $t_{min}$ represent the largest and smallest timestamps, respectively, and $E$ denotes the number of events) compared to other datasets, as discussed by Cong et al. (2023). Specifically, LastFM has an average time gap of 106, whereas Reddit's average time gap is 4, Wiki's average time gap is 17, MOOC's average time gap is 3.6, and GDELT's average time gap is 0.1.

• Consequently, even without staleness in the baseline method, the node memory in the LastFM graph tends to become significantly outdated Rossi et al. (2021), as discussed in Section 3.3. Our staleness mitigation strategy eliminates the outdated node representation by aggregating the memories of the recently active nodes with the highest similarity. This approach helps mitigate the impact of the large time gap present in LastFM datasets, ultimately leading to an improvement in AP compared to the baseline methods.

### E.3.2 CONVERGENCE OF THE JODIE AND APAN.

We further provide the convergence of JODIE and APAN models on five datasets in Fig. 15 and Fig. 16. We can see that the training curves of all models largely overlap with the baselines (TGL and Presample), demonstrating that MSPipe preserves the convergence rate. Notably, MSPipe-S achieves better performance than the other variants on the REDDIT and GDELT datasets.

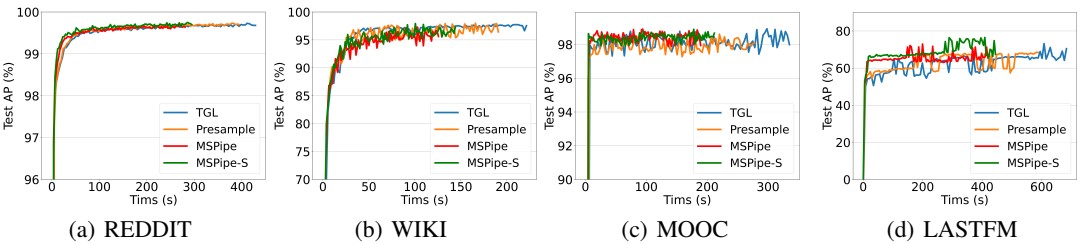

|     (a) REDDIT     |     (b) WIKI     |     (c) MOOC     |     (d) LASTFM     |

Figure 15: Convergence of JODIE training. the x-axis is the wall-clock training time, and the y-axis is the test average pricision

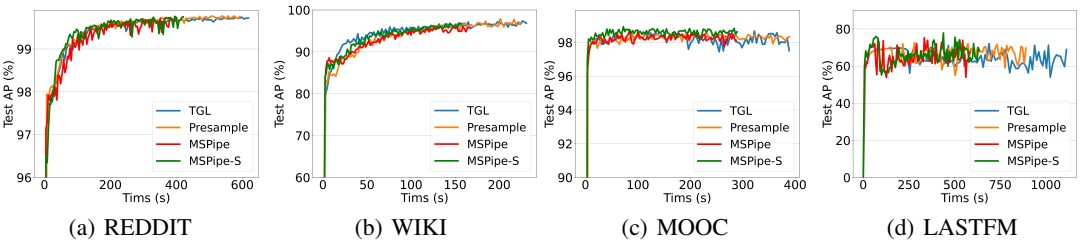

|     (a) REDDIT     |     (b) WIKI     |     (c) MOOC     |     (d) LASTFM     |

Figure 16: Convergence of APAN training. the x-axis is the wall-clock training time, and the y-axis is the test average pricision

### E.3.3 COMPARISON BETWEEN DIFFERENT STALENESS BOUND

Furthermore, we present a comprehensive comparison of various staleness bounds across multiple datasets including REDDIT, WIKI, LASTFM, and GDELT, using the TGN model, in order to validate the efficacy of MSPipe. The results consistently demonstrate that MSPipe outperforms other staleness bound options in terms of both throughput and accuracy across all datasets. As shown in Fig 20, the number of staleness $k_i$ will soon converge to a steady minimal staleness value. To represent this minimal staleness bound, we utilize a fixed value that corresponds to the steady state. This choice allows us to showcase the minimal staleness bound effectively.

### E.3.4 THE DISTRIBUTION OF $\Delta t$ ON OTHER DATASETS.

We introduce $\Delta t$ as the duration since a node $v$'s memory was last updated, which differs from the $\Delta t$ in the TGNN inference system Wang & Mendis (2023); Zhou et al. (2022a). The $\Delta t$ defined in

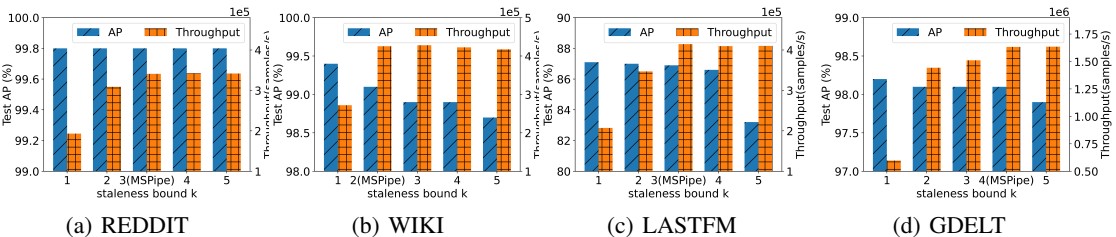

(a) REDDIT   (b) WIKI   (c) LASTFM   (d) GDELT

Figure 17: Staleness error comparison on TGN. MOOC datasets is presented in Fig. 11

TGOpt Wang & Mendis (2023) and Zhou *et al.* Zhou et al. (2022a) are designed for the time-encoder, which is computed by the difference between current events' timestamp and their historical events' timestamps with their neighbors. We further post the distribution of $\Delta t$ of the remaining datasets in Fig. 18 and observed that the $\Delta t$ in all datasets follow the power-law distribution, indicating that most $\Delta t$ values are small and that most node memories are not stale or constant. This observation provides insights into the occurrence patterns of nodes in different dynamic graphs, and our similarity-based staleness mitigation mechanism focuses on compensating for memory vectors with stale $\Delta t$ values in the long tail of the distribution.

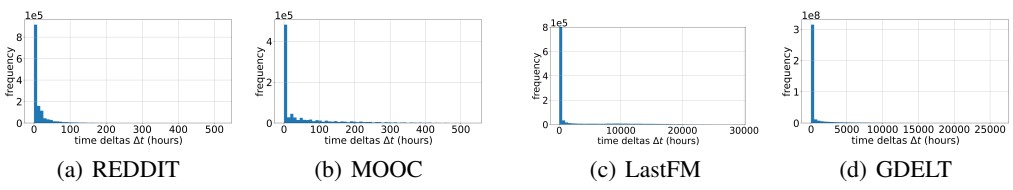

(a) REDDIT   (b) MOOC   (c) LastFM   (d) GDELT

Figure 18: Distribution of $\Delta t$

### E.3.5 ANALYSIS OF THE NODE MEMORY SIMILARITY.

We compensate the stale node memory by finding their most similar and recently active nodes with the intuition that similar nodes have resembling representations that facilitate the stale node to obtain more updated information. The most similar nodes are computed by counting their common neighbors to get Jaccard similarity. As illustrated in Fig. 19, our mechanism for identifying the most recent similar nodes can locate those with representations that are not only similar but also more recently updated than randomly selected nodes. We use cosine similarity as the evaluation metric for similarity.

### E.4 THE VARIANCE OF $k_i$ WITH RESPECT TO $i$

We further evaluate the variance of $k_i$ when the $i$ changes. As shown in Fig 20, the number of staleness $k_i$ will soon converge to a steadily minimal staleness value. This is because of the periodic manner of the GNN training as the computation time of different training stage is quite steady.

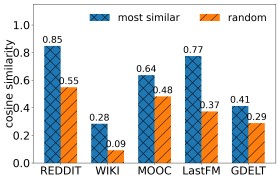

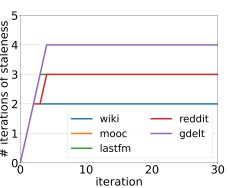

Figure 19: The cosine similarity of the memory vectors between the target nodes (with staled node memory) and their most similar nodes or random nodes

Figure 20: The minimal number of staleness $k_i$ in different iteration $i$

Table 8: Batch size sensitive analysis. The best results are in **bold** second-best are underlined.

| Batch size | Scheme | REDDIT | | WIKI | | MOOC | | LASTFM | | GDELT | |
|---|---|---|---|---|---|---|---|---|---|---|---|
| | | AP(%) | Speedup | AP(%) | Speedup | AP(%) | Speedup | AP(%) | Speedup | AP(%) | Speedup |
| Batch 300 (2000 for GDELT) | TGL | **99.8** | 1× | **99.5** | 1× | **99.4** | 1× | **88.1** | 1× | **98.5** | 1× |
| | Presample | **99.8** | 1.26× | **99.5** | 1.08× | **99.4** | 1.04× | 88.0 | 1.51× | **98.5** | 1.12× |
| | MSPipe | **99.8** | **1.73×** | 99.4 | **1.67×** | **99.4** | **1.47×** | 87.2 | **1.90×** | 98.2 | **1.93×** |
| | MSPipe-S | **99.8** | 1.68× | **99.5** | 1.65× | **99.4** | 1.45× | 88.0 | 1.86× | **98.5** | 1.88× |
| Batch 900 (6000 for GDELT) | TGL | **99.8** | 1× | **98.9** | 1× | **98.6** | 1× | 86.9 | 1× | 97.8 | 1× |
| | Presample | **99.8** | 1.10× | **98.9** | 1.12× | **98.6** | 1.10× | 86.9 | 1.37× | 97.8 | 1.26× |
| | MSPipe | **99.8** | **1.62×** | 98.5 | **1.49×** | **98.6** | **1.58×** | 86.7 | **1.87×** | 97.7 | **2.01×** |
| | MSPipe-S | **99.8** | 1.56× | **98.9** | 1.46× | **98.6** | 1.53× | **87.8** | 1.80× | 98.2 | 1.93× |
| Batch 1200 (8000 for GDELT) | TGL | **99.8** | 1× | **98.5** | 1× | 98.3 | 1× | 85.8 | 1× | 97.1 | 1× |
| | Presample | **99.8** | 1.34× | **98.5** | 1.37× | 98.3 | 1.32× | 85.8 | 1.56 | 97.1 | 1.28× |
| | MSPipe | **99.8** | **1.64×** | **98.5** | **1.48×** | 98.3 | **1.69×** | 85.8 | **1.92×** | 97.1 | **1.99×** |
| | MSPipe-S | **99.8** | 1.59× | **98.5** | 1.45× | **98.8** | 1.62× | **86.2** | 1.84× | **98.1** | 1.90× |
| Batch 1600 | TGL | **99.8** | 1× | **98.4** | 1× | 97.9 | 1× | **84.4** | 1× | | |
| | Presample | **99.8** | 1.38× | **98.4** | 1.39× | 97.9 | 1.33× | **84.4** | 1.51× | | |
| | MSPipe | **99.8** | **1.66×** | 98.1 | **1.58×** | 97.9 | **1.71×** | 82.7 | **1.97×** | | |
| | MSPipe-S | **99.8** | 1.58× | 98.3 | 1.53× | **98.7** | 1.64× | 84.2 | 1.88× | | |

Table 9: MSPipe compares with baseline methods using larger batch size.

| Dataset | AP(%) | REDDIT Time(s) | Speedup | AP(%) | WIKI Time(s) | Speedup | AP(%) | MOOC Time(s) | Speedup | AP(%) | LastFM Time(s) | Speedup |
|---|---|---|---|---|---|---|---|---|---|---|---|---|
| TGL batch 600 | 99.8 | 7.31 | 1× | **99.4** | 2.41 | 1× | **99.4** | 4.31 | 1× | **87.2** | 13.10 | 1× |
| MSPipe batch 600 | 99.8 | **4.14** | **1.77×** | 99.1 | **1.57** | **1.54×** | 99.3 | **2.88** | **1.50×** | 86.9 | **6.55** | **1.87×** |
| TGL batch 900 | 99.8 | 5.22 | 1.40× | 98.9 | 2.03 | 1.19× | 98.7 | 3.18 | 1.36× | 86.9 | 10.10 | 1.30× |
| TGL batch 1200 | 99.8 | 4.48 | 1.63× | 98.5 | 1.83 | 1.32× | 98.3 | 2.99 | 1.44× | 85.8 | 8.43 | 1.55× |

## E.5 BATCH SIZE SENSITIVITY ANALYSIS

To further validate the effectiveness of MSPipe in different batch sizes, we conducted batch size sensitivity evaluations using the following local batch sizes: 300, 900, 1200, and 1600 for the small datasets, and 2000, 6000, and 8000 for the large dataset (used 600 and 4000 in the original experiments).

As demonstrated in Table 8, MSPipe consistently outperforms all baseline methods in varying batch sizes, achieving up to $2.01\times$ speedup without compromising model accuracy. These results further validate the practicality of MSPipe. It is worth noting that for the same dataset, MSPipe tends to exhibit similar speedup among various batch sizes, indicating no direct correlation between batch size and speedup.

## E.6 COMPARE WITH STRAWMAN METHOD: INCREASE BATCH SIZE

We conducted additional empirical comparisons between MSPipe and baseline methods using larger batch sizes. In Table 9, MSPipe consistently outperforms baseline methods with batch sizes increased by $1.5\times$ and $2\times$, achieving speedups of up to 57% and 32% respectively. While the TGN model experiences up to 1.4% accuracy loss with larger batch sizes, MSPipe maintains high accuracy with a maximum accuracy loss of 0.3%. It is worth emphasizing that MSPipe can be applied with larger batch sizes to further boost training throughput as shown in Table 8.

## E.7 MEMORY OVERHEAD ANALYSIS.

1. In MSPipe, we introduce staleness in the memory module to enable the pre-fetching of features and memory in later iterations. However, unlike PipeGCN and Sancus, where staleness is introduced during GNN training, our TGNN training stage doesn't have staleness. Each subgraph is executed sequentially, so no additional hidden states are incurred during GNN computation.

2. The additional memory consumption in MSPipe arises from the prefetched subgraph, which includes node/edge features and memory vectors. We can compute an upper bound for this memory consumption as follows:

- Let the subgraph in each iteration have a batch size of $B$, node and edge feature dimensions of $H$, node memory dimension of $M$, and an introduced staleness bound of $K$. For each graph event, we have a source node, destination node, and neg_sample node, totaling 3 nodes per sample.

- During subgraph sampling, we use the maximum neighbor size of $N = 10$ to compute the memory consumption, which represents an upper bound. Assuming the data format in Float32 (i.e., 4 bytes), the additional subgraph memory consumption is:

$$3 \times 4KB(N+1)(H+M) + 12KB(N+1) = 12KB(N+1)(H+M) + 12KB(N+1)$$

, where the first term represents the feature and memory usage, $(N + 1)$ is the total number of nodes, and $(H + M)$ is the sum of the feature and memory dimensions. The second term represents the node ID usage.

3. Moreover, we conduct empirical experiments on all the models/datasets with the *torch.cuda.memory_summary()* API. The experiment results are listed in Table tables 10 to 12.

- As observed in the Table tables 10 to 12, the additional memory usage from MSPipe strictly remains below our analyzed upper bound.

- Moreover, the additional memory only introduces an average of 47% more consumption compared to TGL methods. It is important to note that the actual additional memory consumption may be even lower than 47% since PyTorch tends to allocate more memory than it will ultimately use.

Table 10: GPU memory usage of TGN model. The 'Addition' row represent the additional memory usage from MSPipe to TGL by introducing staleness. The 'Theory' row represent the upperbound of additional memory usage by introducing staleness.

| Scheme | REDDIT(MB) | WIKI(MB) | MOOC(MB) | LastFM(MB) | GDELT(GB) |
|---|---|---|---|---|---|
| TGL | 348.16 | 202.75 | 312.72 | 264.19 | 8.12 |
| MSPipe | 507.06 | 303.10 | 428.00 | 352.26 | 11.34 |
| Addition | 158.90 | 100.35 | 115.28 | 88.06 | 3.22 |
| Theory | 193.88 | 129.25 | 162.52 | 162.52 | 4.42 |

Table 11: GPU memory usage of JODIE model. The 'Addition' row represent the additional memory usage from MSPipe to TGL by introducing staleness. The 'Theory' row represent the upperbound of additional memory usage by introducing staleness.

| Scheme | REDDIT(MB) | WIKI(MB) | MOOC(MB) | LastFM(MB) | GDELT(GB) |
|---|---|---|---|---|---|
| TGL | 166.86 | 152.77 | 172.77 | 183.25 | 6.62 |
| MSPipe | 278.54 | 238.30 | 286.50 | 266.98 | 9.42 |
| Addtion | 111.68 | 85.54 | 113.73 | 83.73 | 2.8 |
| Theory | 193.88 | 129.25 | 162.52 | 162.52 | 4.42 |

Table 12: GPU memory usage of APAN model. The 'Addition' row represent the additional memory usage from MSPipe to TGL by introducing staleness. The 'Theory' row represent the upperbound of additional memory usage by introducing staleness.

| Scheme | REDDIT(MB) | WIKI(MB) | MOOC(MB) | LastFM(MB) | GDELT(GB) |
|---|---|---|---|---|---|
| TGL | 229.38 | 215.04 | 196.61 | 208.90 | 7.4 |
| MSPipe | 337.92 | 362.50 | 348.16 | 313.34 | 10.17 |
| Addtion | 108.54 | 147.46 | 151.55 | 104.45 | 2.77 |
| Theory | 193.88 | 129.25 | 162.52 | 162.52 | 4.42 |

