# OpenReview forum: "MSPipe: Minimal Staleness Pipeline for Efficient Temporal GNN Training"
_ICLR.cc/2024/Conference — Submitted to ICLR 2024_

### Official Review · Reviewer_a7Lb · 2023-10-29

**Soundness:** 3 good
**Presentation:** 2 fair
**Contribution:** 2 fair
**Rating:** 5
**Confidence:** 3

**Summary:**

This paper proposes a pipeline scheduling framework, MSPipe, for memory-based TGNN training. The authors discuss the minimal number of staleness iterations and utilize the scheduler to delay memory fetching and prevent resource contention. Experiments validate that the proposed method achieves significant speedup with less accuracy degradation.

**Strengths:**

1. This paper proposed a formulation for TGNN training pipeline and discussed the bottlenecks of the memory module and temporal dependencies. They designed a minimal staleness algorithm and lightweight staleness mitigation method for speeding up TGNN training with less accuracy loss. They also analyzed theoretical convergence to prove the robustness of the proposed method.

2. The structure of the paper is clear and easy to follow.

3. The experimental results are quite extensive.

**Weaknesses:**

1. The scale of the figures should be corrected. Especially in Figure 12. And some of the figures are out of text bound.

2. In Experiments, as different datasets have different distributions of \delta t, how can we find an optimal hyperparameter of \lambda? This parameter selection should be discussed.

3. Although the authors discuss the optimization and asynchronous training from previous work, the proposed method is still easy. The contribution seems insufficient.

**Questions:**

Please see above.

---

> ### Author Response · Authors · 2023-11-15
> **Response to Reviewer a7Lb**
>
> Thank you for your detailed review. We would like to address the weakness you raised and clarify any potential misunderstandings.
>
> > Weakness1: correct the scale of figures especially Figure 12.
>
> We appreciate the detailed review. We have fixed all the scale problems in the paper and the latest version of the paper is updated.
>
>
>
> > Weakness2: as different datasets have different distributions of \delta t, how can we find an optimal hyperparameter of \lambda? This parameter selection should be discussed.
>
> We want to modestly point out that we have indeed included a hyperparameter analysis in Section 4.4 of our paper. Due to space limitations, we only include the results of the LastFM dataset. However, we find that all the datasets exhibit a similar trend to the results shown for the LastFM dataset. Therefore, the insights into the optimal selection of λ across different datasets are applicable to most datasets:
>
> - Specifically, selecting a larger value of λ (e.g., λ > 0.8) tends to improve the model performance. This suggests that retaining a greater proportion of the original stale memory representations and applying a smaller amount of mitigation from similar ones is beneficial.
> - On the other hand, choosing a smaller value of λ (e.g., λ < 0.5) can lead to oversmoothing of node memories by other nodes, thereby degrading the model performance.
>
>
>
> > Weakness3: Although the authors discuss the optimization and asynchronous training from previous work, the proposed method is still easy. The contribution seems insufficient
>
> We respectfully disagree with the reviewer’s assessment of the contribution of our study, and would like to put forth a few points to assert that our contribution is sufficient:
>
> - MSPipe is a novel framework specifically designed to accelerate the TGNN training process. To the best of our knowledge, we are the first to address the bottlenecks of the memory module and provide effective system-algorithm co-design methods to accelerate the TGNN training process, as discussed in Section 2. The comprehensive contributions of MSPipe are summarized in Section 1.
> - The recent surge of advancements in TGNNs underscores the significance of TGNN training within the research community. Notably, the superior model accuracy achieved by memory-based TGNNs [1] further demonstrates the significance of our contribution to accelerating the training process.
> - MSPipe offers accelerated model development speed, saving valuable time for researchers and engineers who would otherwise be waiting for lengthy training processes. MSPipe exhibits superior performance, such as a 2.45× speedup, compared to the existing state-of-the-art framework TGL. It surpasses TGL even with more advanced optimizations from the SOTA static GNN framework while maintaining the model's performance.
> - The training paradigms, bottlenecks, challenges, and dependencies in previous asynchronous training works are entirely different from TGNN training (as detailed in Appendix B). Consequently, these methods cannot be directly applied to expedite TGNN training, highlighting the importance of our contribution to the TGNN training field.
>
> In conclusion, considering the significance and distinctive nature of TGNN research, and the lack of exploration in accelerating TGNN training, we firmly believe that our study brings valuable contributions, novelty, and sufficient advancements to the field.
>
> [1] Farimah Poursafaei, et.al. Towards better evaluation for dynamic link prediction. In Neural Information Processing Systems (NeurIPS) Datasets and Benchmarks, 2022.

---

> ### Author Response · Authors · 2023-11-20
> **Gentle reminder of Reviewer a7Lb**
>
> Thank you again for the feedback on our paper. We hope that our responses have addressed your inquiries and concerns. If this is not the case, please inform us and we would be glad to engage in further discussion.

---

### Official Review · Reviewer_Vi4E · 2023-11-01

**Soundness:** 2 fair
**Presentation:** 3 good
**Contribution:** 2 fair
**Rating:** 6
**Confidence:** 4

**Summary:**

This paper, MSPipe, targets a timely problem: acceleration of (distributed) TGN training. MSPipe considers the 'memory update' procedure in the TGN training, which is the main bottleneck of the TGN training acceleration, and proposes two main ideas. In the baseline optimization, MSPipe overlaps the subgraph sampling and feature fetching. On top of it, first, it uses a staleness-based method to break the TGN memory dependency. Additionally, online scheduling minimizes the staleness bound. Second, using the similarity among vertices, it proposes a staleness mitigation method, which reduces the impact of staleness. With overlapping optimization, staleness-based strategy, and staleness mitigation, MSPipe provides a significant speedup from 1.50 to 2.45x.

**Strengths:**

+ Adequately analyzed the training pipeline of TGNN training and accelerated it. While not very novel, this provides a reasonable and well-designed solution.
+ Proposes some staleness mitigation strategy
+ Provides significant throughput gain
+ Various sensitivity studies in the appendix.

**Weaknesses:**

- Novelty is limited.
- Some accuracy results does not make sense.
- There is no discussion on GPU memory usage.
- Baseline subgraph training methods are outdated compared to caching-based subgraph sampling acceleration works (e.g., SALIENT++).

MSPipe provides an adequate training breakdown of TGNN training and targets to overlap the memory update procedure in TGN training. However, the staleness-based methods are widely used in GNN training. Even though MSPipe suggests that those works differ, the core idea is not very different: breaking the dependency, which is popularly used for GNN frameworks and algorithms.
In addition, when using a staleness-based strategy, the GPU memory usage should be reported, but there is a lack of such a discussion. The staleness mitigation method is interesting and valid but needs more details, and most importantly, it shows somewhat nonsense results in the LASTFM dataset. Overall, MSPipe is interesting and efficient, but some points should be addressed.
While MSPipe points out that it differs from staleness-based works such as PipeGCN and Sancus

**Questions:**

- In the LASTFM dataset, why does MSPipe achieve such high accuracy compared to TGL? Does the staleness mitigation strategy can outperform the AP of the baseline TGL?
- Staleness-based strategies require more memory at the expense of the throughput increase. For example, in Fig. 1(c), when breaking the dependency, the intermediate GPU-memory usage may be twice as much more than the baseline training. Could the authors (theoretically) analyze and report the empirical memory usage overhead?
- Recent works (e.g., SALIENT++, MLSys2023) propose caching-based methodologies to minimize the sampler overhead. Is MSPipe still a valid option when using such methods?

---

> ### Author Response · Authors · 2023-11-15
> **Response to Reviewer Vi4E**
>
> Thank you for your detailed review. We would like to address the weakness you raised and clarify any potential misunderstandings.
>
> > Weakness 1: The staleness-based methods are widely used in GNN training. Even though MSPipe suggests that those works differ, the core idea is not very different: breaking the dependency. Address the novelty and difference with static GNN.
>
> We respectfully disagree with the reviewer’s assessment of the novelty of our works and would like to put forth a few points to assert that our study indeed has novelty:
>
> - The concept of 'staleness' has been extensively researched in various domains for many years, and recent publications continue to explore its utility in different settings. All the staleness methods can be thought of as some sort of ‘*breaking the dependency*’. The novelty of many staleness studies comes from analyzing and applying staleness to a particular problem or setting, as different problems and settings introduce different challenges and require specific theoretical considerations.
> - As discussed in Section 2 and Appendix B, there are multiple staleness-based methods applied in static GNN training. However, the training paradigms, bottlenecks, challenges, and dependencies in these static GNN frameworks are entirely different from TGNN training (as detailed in Appendix B). Consequently, these methods cannot be directly applied to accelerate TGNN training. To the best of our knowledge, we are the first to address the bottlenecks of the memory module and provide effective system-algorithm co-design methods to accelerate the TGNN training process, accompanied by detailed theoretical analysis.
>
> In conclusion, considering the nature of staleness research and the limited exploration in TGNN training, we firmly believe that our study brings value and novelty to the TGNN training field.
>
> > Weakness 2, Question1: Accuracy in LastFM. Does the staleness mitigation strategy can outperform the AP of the baseline TGL in LastFM?
>
> The reasons why our staleness mitigation strategy outperforms the AP of the baseline TGL in the LastFM dataset is due to the unique characteristics of the LastFM datasets:
>
> - The LastFM dataset exhibits a larger average time gap ($\frac{t_{max} - t_{min}}{E}$, where $t_{max}$ and $t_{min}$ represent the largest and smallest timestamps, respectively, and $E$ denotes the number of events) compared to other datasets, as discussed by Cong et al. [1]. Specifically, LastFM has an average time gap of 106, whereas Reddit's average time gap is 4, Wiki's average time gap is 17, MOOC's average time gap is 3.6, and GDELT's average time gap is 0.1.
> - Consequently, even without staleness in the baseline method, the node memory in the LastFM graph tends to become significantly outdated [2], as discussed in Section 3.3. Our staleness mitigation strategy eliminates the outdated node representation by aggregating the memories of the recently active nodes with the highest similarity. This approach helps mitigate the impact of the large time gap present in LastFM datasets, ultimately leading to an improvement in AP compared to the baseline methods.
>
> According to your feedback, we have added the analysis of accuracy in LastFM datasets in the Appendix E.3.1, which aims to provide further clarity and enhance the comprehensiveness of our study.
>
> [1] Weilin Cong, et. al. Do we really need complicated model architectures for temporal networks? In Proceedings of International Conference on Learning Representations, 2023.
>
> [2] Emanuele Rossi, et. al. Temporal Graph Networks for Deep Learning on Dynamic Graphs. In Proceedings of International Conference on Learning Representations, 2021.

---

> ### Author Response · Authors · 2023-11-15
> **Response to Reviewer Vi4E**
>
> > Weakness 3, Question2: Analyze the GPU memory usage and report empirical memory usage overhead.
>
> We appreciate the reviewer's inquiry, and we would like to politely clarify that our method does not introduce twice the memory overhead compared to TGL. Here is the theoretical analysis and empirical results supporting this claim:
>
> 1. In MSPipe, we introduce staleness in the memory module to enable the pre-fetching of features and memory in later iterations. However, unlike PipeGCN and Sancus, where staleness is introduced during GNN training, our TGNN training stage doesn’t have staleness. Each subgraph is executed sequentially, so no additional hidden states are incurred during GNN computation.
>
> 2. The additional memory consumption in MSPipe arises from the prefetched subgraph, which includes node/edge features and memory vectors. We can compute an upper bound for this memory consumption as follows:
>
>    - Let the subgraph in each iteration have a batch size of $B$, node and edge feature dimensions of $H$, node memory dimension of $M$, and an introduced staleness bound of $K$. For each graph event, we have a source node, destination node, and neg_sample node, totaling 3 nodes per sample.
>    - During subgraph sampling, we use the maximum neighbor size of $N=10$ to compute the memory consumption, which represents an upper bound. Assuming the data format in Float32 (i.e., 4 bytes), the additional subgraph memory consumption is: $$3 \times 4 KB(N+1)(H+M) + 12 KB(N+1) = 12KB(N+1)(H+M) + 12KB(N+1)$$ , where the first term represents the feature and memory usage, $(N+1)$ is the total number of nodes, and $(H+M)$ is the sum of the feature and memory dimensions. The second term represents the node ID usage.
>
> 3. Moreover, we conduct empirical experiments on all the models/datasets with the`
>    torch.cuda.memory_summary()`API. The experiment results are listed in the tables blow the text  and we added them in Table 10-12 in Appendix E.7.
>
>    - As observed in the Table 10-12, the additional memory usage from MSPipe strictly remains below our analyzed upper bound.
>
>    - Moreover, the additional memory only introduces an average of 47% more consumption compared to TGL methods. It is important to note that the actual additional memory consumption may be even lower than 47% since PyTorch tends to allocate more memory than it will ultimately use.
>
> Based on the above analysis, our findings indicate that MSPipe effectively accelerates training performance while only introducing a manageable memory overhead.
>
> ***
> **Tables:** GPU memory usage of different models.
> - The 'TGL' and 'MSPipe' rows represent the memory usage from TGL and MSPipe respectively under the same experiment settings in the paper.
> - The 'Addition' row represents the additional memory usage from MSPipe to TGL by introducing staleness.
> - The 'Theory' row represents the upper bound of additional memory usage by introducing staleness.
>
> **TGN model**
>
> | Scheme  | REDDIT(MB) | WIKI(MB) | MOOC(MB) | LastFM(MB) | GDELT(GB) |
> | ------- | ---------- | -------- | -------- | ---------- | --------- |
> | TGL     | 348.16     | 202.75   | 312.72   | 264.19     | 8.12      |
> | MSPipe  | 507.06     | 303.10   | 428.00   | 352.26     | 11.34     |
> | Addtion | 158.90     | 100.35   | 115.28   | 88.06      | 3.22      |
> | Theory  | 193.88     | 129.25   | 162.52   | 162.52     | 4.42      |
>
> **JODIE model**
>
> | Scheme  | REDDIT(MB) | WIKI(MB) | MOOC(MB) | LastFM(MB) | GDELT(GB) |
> | ------- | ---------- | -------- | -------- | ---------- | --------- |
> | TGL     | 166.86     | 152.77   | 172.77   | 183.25     | 6.62      |
> | MSPipe  | 278.54     | 238.30   | 286.50   | 266.98     | 9.42      |
> | Addtion | 111.68     | 85.54    | 113.73   | 83.73      | 2.8       |
> | Theory  | 193.88     | 129.25   | 162.52   | 162.52     | 4.42      |
>
> **APAN model**
>
> | Scheme  | REDDIT(MB) | WIKI(MB) | MOOC(MB) | LastFM(MB) | GDELT(GB) |
> | ------- | ---------- | -------- | -------- | ---------- | --------- |
> | TGL     | 229.38     | 215.04   | 196.61   | 208.90     | 7.4       |
> | MSPipe  | 337.92     | 292.50   | 308.16   | 313.34     | 10.17     |
> | Addtion | 108.54     | 77.46    | 111.55   | 104.45     | 2.77      |
> | Theory  | 193.88     | 129.25   | 162.52   | 162.52     | 4.42      |

---

> ### Author Response · Authors · 2023-11-15
> **Response to Reviewer Vi4E**
>
> > Weakness 4, Question3: Is MSPipe still a valid option when using SALIENT++?
>
> We would like to assert that MSPipe is still a valid option for TGNN training even considering the usage of SALIENT++, which is a follow-up work to SALIENT (used in our experiment):
>
> 1. SAILENT++ focuses on reducing the communication overhead caused by partitioned vertex feature storage by using a feature cache. However, in our experiments, neither TGL nor MSPipe employs partitioned vertex feature storage since the feature storage requirements of all the datasets could be accommodated in the CPU's main memory. Therefore, there was no need to apply partitioned vertex features, and comparing our approach with SALIENT is sufficient.
> 2. The design of SALIENT++ is not applicable to the memory module, which serves as the main bottleneck in TGNN training, even if we were to partition the memory module to fit its setting. SALIENT++ introduces a cache for feature fetching while the feature storage in the main memory will not be updated throughout the training. However, in the case of TGNN training, the memory module needs to be updated every iteration. Consequently, it is crucial to maintain strict consistency between the node memory cache on different GPUs and the memory module. However, SALIENT++ does not address this issue.
> 3. SALIENT++ specifically designs a feature cache for the multi-hop subgraph sampler to reduce communication overhead arising from the neighborhood explosion problem in multi-layer GNNs. However, as discussed in our paper, memory-based TGNN utilizes a single-layer structure. In this case, the main bottlenecks stem from the memory module rather than subgraph sampling and feature fetching, thereby reducing the potential benefits of SALIENT++.
>
> Based on the above reasons, SALIENT++ mainly focuses on different settings that do not address the main challenges in TGNN training, and it is not appropriate to compare MSPipe with SALIENT++. Instead, we compare our approach to SALIENT.

---

> > ### Comment · Reviewer_Vi4E · 2023-11-19
> > **Thanks for the response**
> >
> > Thank you for the detailed response.
> > * On the novelty, I think I am already on the same page with the authors.
> > Indeed, analyzing the given problem to apply the staleness is meaningful.
> > I agree that it is novel to some degree, but not greatly.
> > Given a large body of previous and concurrent work on GNN subproblem + staleness, the proposed work is more of a timely one, rather than being very novel.
> > * Thanks for providing the theoretical and empirical memory usage. The additional memory usage is less than what I thought, but still a huge overhead, especially for modern GNNs where the memory capacity is often the system bottleneck. If the paper is accepted, I hope the authors place this in the main body, not the appendix.
> > * The discussion for the LastFM data lacks enough support, but seems to make much sense.
> >
> > Considering that the response addresses much of the initial concerns, I have raised my rating to 6.

---

> > > ### Author Response · Authors · 2023-11-21
> > > **Thanks for the prompt feedback**
> > >
> > > Thanks for your detailed review and really appreciate your reply! We would like to further address your responses:
> > >
> > > - We promise that, if our paper is accepted, we will include the detailed memory consumption analysis in the main body.
> > > - We sincerely appreciate the reviewer's recognition of our contributions and we would like to emphasize that in addition to addressing the unique challenges in TGNN training, we have also tackled an unsolved problem in the GNN subproblem + staleness domain from two key aspects: 1) we strategically introduce a minimal staleness bound, in contrast to previous GNN staleness works [1, 2] that rely on randomly selected bounds by human discretion, and 2) we have proposed a novel regularization method specifically designed to mitigate the impact of staleness.
> > > - We thank the reviewer for acknowledging that our discussion is meaningful. We would like to further address this point by highlighting that, even though [3] also discusses the accuracy improvements achieved through their methods in other aspects, strictly explaining the accuracy gain resulting from different TGNN training methods may still be an open question.
> > >
> > > Thanks again for your prompt feedback.
> > >
> > > [1] Wan, et al. PipeGCN: Efficient full-graph training of graph convolutional networks with pipelined feature communication. In The Tenth International Conference on Learning Representations, 2022.
> > >
> > > [2] Peng, et al. Sancus: staleness-aware communication-avoiding full-graph decentralized training in large-scale graph neural networks. Proceedings of the VLDB Endowment, 15(9):1937–1950, 2022.
> > >
> > > [3] Weilin Cong, et. al. Do we really need complicated model architectures for temporal networks? In Proceedings of International Conference on Learning Representations, 2023.

---

### Official Review · Reviewer_juas · 2023-11-01

**Soundness:** 3 good
**Presentation:** 3 good
**Contribution:** 3 good
**Rating:** 6
**Confidence:** 3

**Summary:**

In the training process of memory-based TGNN, memory modules are used to store the temporal information computed by the RNN. These vectors, once computed on the GPU, are stored in the CPU memory, which introduces significant overhead, resulting in underutilization of the GPU. This work introduces staleness into the memory modules to break the time dependency, achieved through the minimal staleness algorithm. The algorithm determines the minimal staleness bound, denoted as 'k'. During the computation at the current i-th iteration, the results from the (i-k)-th iteration are used instead of the (i-1) iteration's results, allowing the training phases to be pipelined. This enables the GPU to seamlessly execute computations without waiting for data preparation, maximizing the TGNN training throughput. Additionally, this work proposes a similarity-based staleness mitigation method to further enhance the model's accuracy.

**Strengths:**

1. Overall, the two optimization methods are reasonable.
2. The experiment result is promising. It can be observed that the first optimization method, introducing staleness to break temporal dependencies, can improve training throughput and acceleration ratio. Furthermore, the algorithm identifies the minimal staleness bound 'k,' and experiments confirm its optimality in the trade-off between accuracy and throughput. The second optimization method, introducing a staleness mitigation approach, can enhance the model's precision.
3. The method is novel. Inspired by PipeGCN's breakthrough in breaking the inter-layer dependencies of GNN, this work introduces, for the first time, a method to break the time dependencies of memory modules during TGNN training and provides detailed theoretical derivations.

**Weaknesses:**

1. Section 3.2 "Minimal-staleness bound k" should be the main contribution of this work, but the presentation is unclear. The process of determining the minimum k involves presenting three formulas corresponding to three constraints. The rationale behind the first two formulas is questionable, and it is not explained why these formulas satisfy the constraints.
2. When conducting ablation studies, increasing the influence of GPU samplers is necessary, as TGL uses a CPU sampler, while MSPipe employs a GPU sampler. In addition to the four scenarios in Table 2, it is necessary to add scenarios where MSPipe uses a CPU sampler.

**Questions:**

1. In Section 3.2, titled "Resource-aware online pipeline schedule", it discusses pipeline scheduling after determining the Minimal-staleness bound, denoted as 'k.' In this section, Figure 6 is referenced for illustration. However, Figure 6(a) clearly does not satisfy the formula for determining the minimum 'ki' as outlined in Equation 1. Nevertheless, it does satisfy the constraints mentioned in the text, highlighting a contradiction between the formula and the stated constraints.
2. The figure numbering is disordered: Fig. 7 appears before Fig. 6 in the text. The same figures appear multiple times: Fig. 4 (a) and Fig. 6 (a).
3. The typo in Eqn.5: $j$ in second line should be $j+1$.

---

> ### Author Response · Authors · 2023-11-15
> **Response to Reviewer juas**
>
> Thank you for your detailed review. We would like to address the weakness you raised and clarify any potential misunderstandings.
>
> > Question 1: Figure 6(a) does not satisfy the formula for determining the minimum 'ki' but it satisfied the constrained in Eqaution5.
>
> We want to clarify a potential misunderstanding of Figure 6(a):
>
> - You are right that Figure 6(a) does not represent the minimum '$k_i$' value and Figure 6(b) shows the minimum '$k_i$'  by our minimal staleness optimization. The purpose of including both Figure 6(a) and Figure 6(b) was to compare the pipeline schedule with different minimum '$k_i$'  values. Figure 6(a) serves as a baseline, while Figure 6(b) illustrates how the memory fetching stage can be delayed (indicated by the green dashed line) when applying the minimal staleness '$k_i$'  optimization.
>
> To address the reviewer's concern, we have updated the latest version of the submission to include only Figure 6(b), removing Figure 6(a) to avoid any confusion.
>
>
>
> > Weakness 1: The rationale behind the first two formulas is questionable, and it is not explained why these formulas satisfy the constraints.
>
> We would like to assert that our formulas and constraints are reasonable:
>
> 1. The explanation of these formulas and constraints can be found in Section 3.2. The first two formulations represent the start time of the sample stage and feature fetching stage. The sample stage, being the initial stage, can start immediately after the completion of the sample stage in the previous iteration. For the feature fetching stage, as feature fetching and memory fetching contents for PCIe bandwidth, therefore, the feature fetching stage needs to wait for the completion of the memory fetching stage. For the other stages, there’s no resource contention between different stages so this is why the first two formulas differ from the other stages (the third formula) in our approach.
> 2. **Why the formulation in Equation 5 is reasonable:** We first compute the starting time and end time of different stages in different iterations. The computed starting time represents the earliest possible start time for each stage. As discussed in the paper, Our aim was to parallelize the stages between iterations to maximize throughput while considering two constraints: avoiding resource competition and preventing simultaneous execution of the same stage from different iterations. Figure 4(a) illustrates how the starting time of each stage adheres to these constraints. As there is plenty of bubble time between stages, it may be possible to delay the execution of some stages to obtain a smaller staleness bound.
> 3. **Why the formulations in Equation 5 satisfy the constraints:** Based on the starting time and end time obtained from Equation 5, we iterate through the stages in different iterations to identify opportunities for execution delay, which can lead to a smaller staleness bound. As discussed in the paper, the first constraint aims to delay the memory fetch stage until the memory update stage from $k_i$ iterations has finished. Simultaneously, the second constraint ensures that the delayed memory fetch stage does not impede the subsequent GNN training stage, allowing uninterrupted execution of GNN training stages on the GPU.
>
> We hope this clarification addresses your concerns and provides a clearer understanding of the rationale behind the first two formulas and how they satisfy the constraints outlined in our approach.
>
>
>
> > Weakness 2: Increasing the influence of GPU samplers is necessary.
>
> The impact of the GPU sampler is relatively minor compared to the gains achieved through our pipeline mechanism and minimal staleness scheduling.
>
> - We want to point out that we have already included a thorough analysis of the influence of GPU sampler in Table 5 in Appendix C.3.
> - Table 5 shows that our sampler is 24.3% faster than TGL’s CPU sampler for 1-hop most recent sampling, which accounts for only 3.6% of the total training time.
>
> Therefore, the performance gain is primarily attributed to our pipeline mechanism and resource-aware minimal staleness schedule but not to the acceleration of the sampler.
>
>
>
> > Question 2: Fig. 7 appears before Fig. 6 in the text. The same figures appear multiple times: Fig. 4 (a) and Fig. 6 (a):
>
> Thanks for pointing out the format problem. As suggested, we have fixed the figure order and deleted Figure 6(a) to avoid misunderstandings.
>
>
> > Question 3: Typo in Eqn5
>
> Thanks for your detailed review. We have fixed this typo in the latest version.

---

> ### Author Response · Authors · 2023-11-20
> **Gentle reminder of Reviewer juas**
>
> Thank you again for the feedback on our paper. We hope that our responses have addressed your inquiries and concerns. If this is not the case, please inform us and we would be glad to engage in further discussion.

---

### Official Review · Reviewer_6Ut8 · 2023-11-08

**Soundness:** 1 poor
**Presentation:** 3 good
**Contribution:** 1 poor
**Rating:** 1
**Confidence:** 4

**Summary:**

Memory based TGNNs are an important subclass of TGNNs that rely on message passing to update node memory between events. However message updates suffer from a staleness problem. Since temporal edges are used as ground truth in self-supervised TGNNs, updates to node memory need to be delayed to avoid the information leak problem i.e., the updated memory of a node cannot be utilized for training during the current batch, instead the memory updates are applied at the end of each training iteration. Thus memory based TGNNs have temporal dependencies which affect training accuracy. In order to solve this problem, the authors propose a TGNN training framework called MSPipe which consists of a minimal staleness algorithm that 1) schedules the training pipeline by satisfying a minimal staleness bound condition and (2) exploits a  staleness mitigation method that leverages the memories of recently updated nodes with the highest similarity in order to reduce the staleness error. They provide experimental results comparing MSPipe  to existing TGNN  frameworks TGL and SALIENT.

**Strengths:**

The paper formalizes the pipeline for memory-based TGNN training and proposes a staleness aware algorithm that ensures efficient training while minimizing the memory staleness bound.

Experimental results show good runtime speedup with little decrease in accuracy.

**Weaknesses:**

The prime motivations behind this paper (eliminating staleness while improving training time)  are not valid.

--- The paper proposes a pipeline scheduling framework to improve the runtime of TGL. As mentioned by the authors, the main factor that leads to inefficient TGN training is the dependency on the execution order of memory fetching and updating. The assumption that the memory update should be applied at the end of each training iteration is not valid. As shown in Fig. 2, both memory update and GNN training can naively be executed in parallel. The cost of a memory update can easily be overlapped with (absorbed by) GNN training, as GNN training is the main overhead. Therefore, there is no need to use stale memory.

--This work is to improve previous work TGL. There seems to be a major design flaw.  They proposed to fetch a stale version of the node memory to overlap part of the mini-batch generation overhead with the actual training. However, there’s no need to use stale memory at all, because updated node memory is firstly computed in the previous GNN training iteration, which can be directly used in the next iteration. The sampler can simply include the information of “which node memory should be fetched from the global pool and which node memory should be used as in previous iteration” in the mini-batch data.

Experiments require improvements for soundness.

-- The runtime breakdown shown in Table 1 seems doubtful. The sample overhead is larger than expected, considering that the authors have implemented a GPU-based most recent neighbor sampler and only one-hop neighbors are required for each node.
-- MSPipe calculates a minimal staleness bound $k_i $ for each iteration $i$. However, there is a lack of experiments that demonstrate the variation of $k_i$ with respect to $i$.
-- Fig. 11 depicts a fixed staleness bound value derived by MSPipe for one dataset, which can be confusing.

**Questions:**

This work is to improve previous work TGL. There seems to be a major design flaw.  They propose to fetch a stale version of the node memory to overlap part of the mini-batch generation overhead with the actual training. However, there’s no need to use stale memory at all, because updated node memory is firstly computed in the previous GNN training iteration, which can be directly used in the next iteration. The sampler can simply include the information of “which node memory should be fetched from the global pool and which node memory should be used as in previous iteration” in the mini-batch data.

---

> ### Author Response · Authors · 2023-11-15
> **Response to Reviewer 6Ut8**
>
> Thank you for your detailed review. However, we would like to address the weakness you raised and clarify any potential misunderstandings.
>
>
>
> > Weakness 1: “The cost of a memory update can easily be overlapped with (absorbed by) GNN training, as GNN training is the main overhead. Therefore, there is no need to use stale memory.”
>
> We respectfully disagree with the statement and would like to assert that the cost of a memory update (as the main overhead)  can not be overlapped by TGNN training. We highlight two reasons why it is  infeasible to hide the memory update overhead with TGNN training:
>
> 1. TGNN training stage cannot fully overlap with the memory update stage because the memory update stage can only be applied after the memory updater updates the memory within the TGNN training stage, as discussed in Section 2. Furthermore, the computation overhead of the memory updater may be larger than the embedding modules [1]. Consequently, the available space for the memory update stage to overlap with the TGNN training stage becomes even smaller.
> 2. We have already overlapped the memory update stage as much as possible with the TGNN training stage. TGNN training can be decomposed into three steps in total: memory updater computes the updated memory -> TGNN layer computes the embeddings -> get loss and backward (including all-reduce). The latter two stages can indeed be parallelized with the memory update stage and this is what we already implement in our experiment, which is aligned with TGL[2]. However, even with these overlaps, the memory update stage still takes up to 31.7% of the time indicated in Table 1, making it impossible to completely hide the overhead.
>
> Therefore, our design to utilize stale memory is necessary in order to effectively hide the overhead resulting from the memory module.
>
> [1] Xuhong Wang, et al. Apan: Asynchronous propagation attention network for real-time temporal graph embedding. In Proceedings of the 2021 international conference on management of data.
>
> [2] Hongkuan Zhou, et a.Tgl: A general framework for temporal gnn training on billion-scale graphs. VLDB 2022.
>
>
>
> > Weakness2: “Design flaw. The sampler can simply include the information of “which node memory should be fetched from the global pool and which node memory should be used as in previous iteration” in the mini-batch data.”
>
> We would like to point out that the suggested design, despite being conceptually plausible, is not feasible due to technical constraints in a distributed training system. Therefore, it should not be considered as evidence to deny our contributions. Here are the detailed reasons:
>
> 1. During distributed training, the updated node memories are distributed among various GPUs. In order to determine and fetch the desired node memory for the next iteration, additional communication overhead would be required to find the corresponding memory residing on another GPU. This would introduce unnecessary complexity and hinder the efficiency of the training process.
> 2. Based on the memory update algorithm in TGN, the same target node is involved multiple times in a training batch and receives multiple updated node memories. The final updated memory is chosen based on either the most recent update or the mean update. If we don't write back to the CPU memory and compute the most recent or mean value, it would introduce more overhead for the distributed GPUs to synchronize with each other and determine the most recent or mean value. This would negatively impact the overall training performance, making it impossible to achieve the same throughput as MSPipe which can seamlessly execute the TGNN training stage.
> 3. In addition to the overhead mentioned above, the node memory that will be used in the next iteration still needs to be updated to the memory stored in the CPU main memory, otherwise, the global memory store will get outdated very soon. The presence of unnecessary node memories in the GPU's memory would introduce extra memory overhead when they are no longer needed in future iterations.

---

> ### Author Response · Authors · 2023-11-15
> **Response to Reviewer 6Ut8**
>
> > Weakness 3: “runtime breakdown for sample is larger than expected.”
>
> We would like to assert that the runtime breakdown of MSPipe is indeed reasonable:
>
> 1. We have thoroughly addressed the effect of the GPU sampler in our submission, providing a detailed analysis of our GPU sampler and a comprehensive time breakdown comparison with TGL's CPU sampler. We found that our sampler is 24.3% faster than TGL’s CPU sampler for 1-hop most recent sampling, which accounts for only 3.6% of the total training time. Therefore, the performance gain is primarily attributed to our pipeline mechanism and resource-aware minimal staleness schedule but not to the acceleration of the sampler. These findings are presented in Appendix C.3 and Table 5.
> 2. The temporal sampler needs an additional operation to follow the temporal order during sampling than the static GNN sampler. Therefore it may have a larger overhead than a static GNN sampler. Moreover, comparing the sampling overhead with a static GNN sampler is unnecessary in MSPipe.
>
>
>
> > Weakness 4: “demonstrate the variation of ki with respect to i”
>
> We understand the desire for a comprehensive experiment. We have included an experiment about the $k_i$ with respect to $i$ in Figure 20 in Appendix E.4. As we can see in Figure 20, the number of staleness $k_i$ will soon converge to a steadily minimal staleness value. This is because of the periodic manner of the GNN training as the computation time of different training stages is quite steady.
>
>
>
> > Weakness 5: “Fig. 11 depicts a fixed staleness bound value derived by MSPipe for one dataset, which can be confusing.”
>
> We apologize for the confusion in Figure 11 and please allow us to clarify:
>
> 1. The purpose of Figure 11 was to highlight that MSPipe is capable of identifying the minimal staleness bound that achieves the highest training throughput without compromising model accuracy, surpassing random selection. Due to the space limit, we only presented the MOOC dataset in the main section, while the results for all other datasets were included in Figure 17 in Appendix E3.3.
> 2. As explained before, the number of staleness $k_i$ will soon converge to a steady minimal staleness value. To represent this minimal staleness bound, we utilize a fixed value that corresponds to the steady state. This choice allows us to showcase the minimal staleness bound effectively.
>
> According to your feedback, we have modified the presentation of the experiment to make the purpose of Figure 11 more clear.

---

> ### Author Response · Authors · 2023-11-20
> **Gentle reminder to Reviewer 6Ut8**
>
> Thank you again for the feedback on our paper. We hope that our responses have addressed your inquiries and concerns. If this is not the case, please inform us and we would be glad to engage in further discussion.

---

### Author Response · Authors · 2023-11-15
**Summary of the updates in the Paper**

In response to the reviewers' feedback, we have incorporated the following updates in the revised paper:

- We corrected the scale of Figure 12 and fixed the other figures to make them inside the text box and we adjusted the order of Figure 6 and Figure 7.
- We included an experimental result about the number of staleness $k_i$ with respect to iteration $i$ in Fig 20 in Appendix E.4.
- We added the theoretical analysis and experimental results about the GPU memory usage of the baseline and MSPipe methods in Table 10-12 in Appendix E.7.
- We removed Figure 6(a) and only left Figure 6(b) in Figure 6 to avoid confusion.
- We fixed the typo in the second line of Equation 5 from $j$ to $j+1$.
- We fixed some presentation issues pointed out by the reviewer in Section 4.3.
- We add ‘the choice of staleness bound’ in Appendix B to further distinguish MSPipe from other static GNN frameworks.
- We add the accuracy analysis of LastFM datasets in Appendix E.3.1 to provide further clarity.

---

### Meta-Review · Area_Chair_vs5S · 2023-12-06

**Metareview:**

Summary:

Strengths:
+ The paper formalizes the pipeline for memory-based TGNN training and proposes a staleness aware algorithm that ensures efficient training while minimizing the memory staleness bound.
+ Experimental results show good runtime speedup with little decrease in accuracy.
+ The experiment result is promising. It can be observed that the first optimization method, introducing staleness to break temporal dependencies, can improve training throughput and acceleration ratio. Furthermore, the algorithm identifies the minimal staleness bound 'k,' and experiments confirm its optimality in the trade-off between accuracy and throughput. The second optimization method, introducing a staleness mitigation approach, can enhance the model's precision.
+ The method is novel. Inspired by PipeGCN's breakthrough in breaking the inter-layer dependencies of GNN, this work introduces, for the first time, a method to break the time dependencies of memory modules during TGNN training and provides detailed theoretical derivations.

Weaknesses:
- One reviewer believes that there is no need to use stale memory.
- There seems to be a major design flaw (see corresponding review).
- Experiments require improvements for soundness.
- The runtime breakdown shown in Table 1 seems doubtful.
- There is a lack of experiments that demonstrate the variation stated. one dataset, which can be confusing.

**Justification For Why Not Higher Score:**

For the reasons above

**Justification For Why Not Lower Score:**

N/A

---

### Decision · Program_Chairs · 2024-01-16

Reject